# Repeated exposure with short-term behavioral stress resolves pre-existing stress-induced depressive-like behavior in mice

Eun-Hwa Lee [1], Jin-Young Park [1], Hye-Jin Kwon[1] & Pyung-Lim Han [1,2]✉

Chronic stress induces adaptive changes in the brain via the cumulative action of glucocorticoids, which is associated with mood disorders. Here we show that repeated daily five-minute restraint resolves pre-existing stress-induced depressive-like behavior in mice. Repeated injection of glucocorticoids in low doses mimics the anti-depressive effects of short-term stress. Repeated exposure to short-term stress and injection of glucocorticoids activate neurons in largely overlapping regions of the brain, as shown by c-Fos staining, and reverse distinct stress-induced gene expression profiles. Chemogenetic inhibition of neurons in the prelimbic cortex projecting to the nucleus accumbens, basolateral amygdala, or bed nucleus of the stria terminalis results in anti-depressive effects similarly to short-term stress exposure, while only inhibition of neurons in the prelimbic cortex projecting to the bed nucleus of the stria terminalis rescues defective glucocorticoid release. In summary, we show that short-term stress can reverse adaptively altered stress gains and resolve stress-induced depressive-like behavior.

[1] Department of Brain and Cognitive Sciences, Scranton College, Ewha Womans University, Seoul 03760, Republic of Korea. [2] Department of Chemistry and Nano Science, College of Natural Science, Ewha Womans University, Seoul 03760, Republic of Korea. ✉email: plhan@ewha.ac.kr

Stress has been described as "the nonspecific response of the body to any demand for change"[1,2]. The stress response proceeds by activating the hypothalamic-pituitary-adrenal (HPA) axis, which causes a release of glucocorticoids (GCs, cortisol in humans and corticosterone in rodents) from the adrenal glands[3,4]. GCs normally stimulate energy metabolism and general physiological activity and increase vigilance, memory, and other cognitive functions[5,6]. Therefore, normal physiological stress has necessary and beneficial effects in daily life. From another point of view, basal blood GC levels are low, but they exhibit circadian oscillation, with the highest levels in the early morning and the lowest levels at midnight. The daily fluctuation of GCs might be important for resetting the stress coping system, but the biological significance of oscillating low GC levels has not been carefully investigated.

Chronic stress is a potent risk factor for various psychiatric illnesses, including major depressive disorder[7,8]. Chronic stress produces genomic and neuronal changes in various brain regions beyond their homeostatic capability[4,6,8]. One of the brain regions that undergo functional and structural changes following chronic stress is the medial prefrontal cortex (mPFC) in rodents[9–11]. Chronic stress reduces glutamate transmission and related signaling events[12,13], decreases neuronal activity[13], and causes dendritic atrophy and spine loss in the mPFC[9], which leads to persistent depressive behavior. Furthermore, the reduced neuronal activity in the mPFC induced by chronic stress leads to dysregulation of the HPA axis[4,10,14]. These results suggest that GC-dependent changes in the mPFC are critical for stress-induced depressive behavior. However, several studies have reported that glucocorticoid receptor (GR) agonists produce acute antidepressant effects. Dexamethasone (a synthetic glucocorticoid) treatment in conjunction with sertraline/fluoxetine for 4 days[15] or a single intravenous injection of cortisol[16,17] acutely improves depressive symptoms in patients with depression. Low-dose corticosterone administered for 4 days in mice reduces immobility time in the forced swim test[18]. These results raise the possibility that GCs can exert anti-depressive effects under certain conditions. Thus, whether and if so how GCs exert pro-depressive or anti-depressive effects has been a long-standing unsolved problem[19]. Chronic stress produces adaptive changes via the cumulative action of GCs. If existing stress gains are cumulatively strengthened by chronic stress, prior stress gains should be in a transiently unstable state to integrate the incoming stress inputs. This idea also raises the question that implementing short-term stress could change existing stress gains.

Behavioral appraisals are used to treat the emotional dysfunction of psychiatric disorders[20]. Cognitive reappraisal is a type of behavioral appraisal that regulates emotion by reinterpreting an emotion-provoking situation or reframing emotional expression[21,22]. A related but more complex form of behavioral appraisals includes cognitive behavioral therapy (CBT)[23,24] and exposure therapy[25], which involve stress responses. Although the behavioral appraisals are beneficial for post-traumatic stress disorder[25,26] and depression[23,27], the underlying neural mechanisms remain largely unknown, and it remains unclear if stress responses are required for therapeutic effects of the behavioral methods or they are an impediment to be properly controlled.

In this study, we test the hypothesis that a behavioral method implementing short-term stress could provide an opportunity to modify preexisting stress gains instead of strengthening them, and demonstrate that a behavioral method repeatedly implementing short-term stress can resolve preexisting adaptively altered stress gains in the mPFC and rescue stress-induced depressive behavior.

## Results

**Repeated short-term stress produced anti-depressive effects.** We tested the hypothesis that short-term behavioral stress would not strengthen existing stress gains but resolve them by applying repeated brief restraint in stress-induced models of depression. Mice (C57BL/6) treated with daily 2-h restraint for 14 days (chronic restraint stress, CRST) showed reduced sociability in the two-chamber social interaction test (SIT), decreased sucrose preference in the two-bottle sucrose preference test (SPT), and increased immobility in the tail suspension test (TST) and forced swim test (FST) (Fig. 1a–e). We found that repeated stress induced by daily 5-min restraint for 14 days (RS5), which in itself did not produce depressive effects, reversed the stress-induced behavioral deficits in CRST mice, as did the antidepressant imipramine (Fig. 1b–e). Interestingly, however, treatment with daily 10-min or 15-min restraint for 14 days (RS10 and RS15, respectively) did not produce those effects (Fig. 1b–e). K-means clustering, an unsupervised machine-learning algorithm that groups multiple factors into featured clusters, of the behavioral responses of all individuals, in conjunction with principal component analysis (PCA), yielded two clusters that contained the control or CRST groups in the SIT × SPT × [TST × FST] matrix (Fig. 1f). The RS5- and imipramine-treated individuals were distributed in the cluster containing most of the control animals, whereas the RS10- and RS15-treated animals were in the cluster containing most of the CRST mice (Fig. 1f, g), suggesting that RS5 treatment completely reversed the CRST-induced changes at the individual level, whereas RS10 and RS15 did not. Furthermore, the anti-depressive effects of RS5 were stably maintained for 1 month (Fig. 1h, i). Our dose-ranging study indicated that treatment with the 5-min restraint for 7 days or more, but not 3 or 5 days, was required to produce anti-depressive effects (Supplementary Fig. 1a–f).

Next, we examined whether RS5 treatment was effective in the chronic social defeat stress (CSDS) model, which is unrelated to restraint stress (Fig. 1j). Mice subjected to CSDS were separated into susceptible and resilient subgroups on the basis of the sociability test on day 11 (Supplementary Fig. 2). CSDS-susceptible mice were treated with RS5 for 14 days (Fig. 1j). While CSDS-susceptible mice exhibited behavioral deficits in the SIT, SPT, TST, and FST, CSDS-susceptible mice treated with RS5 exhibited control-like behavior in those tests (Fig. 1k–n). These results suggest that RS5 effects might involve the neural mechanism beyond a simple cognitive reinterpretation of the previous restraint context.

We also examined whether RS5 could produce behavioral changes in ICR mice, an outbred line. A group of ICR mice were subjected to maternal stress during pregnancy (that is, embryonic days E8.5 to E20.5). The progenies of the normal (N) and maternal stress (MS) groups were randomly allocated to receive CRST or CRST + RS5 (Supplementary Fig. 3a, b). CRST treatment in the N and MS groups produced decreased sociability in the SIT, reduced sucrose preference in the SPT, and increased immobility in the TST and FST. RS5 treatment of the CRST-treated N and MS groups reversed their stress-induced behavioral deficits (Supplementary Fig. 3c–f). The results of K-means clustering of individual animals in a matrix of the four independent behavior tests indicated that RS5 treatment shifted CRST-treated N and MS individuals to locate in the cluster containing the normal control (Supplementary Fig. 3g, h). Together, these results suggest that RS5 treatment produces anti-depressive effects in CRST-treated ICR mice and that RS5 can modify depressive-like phenotypes induced by CRST in mice born of stressed mothers during pregnancy.

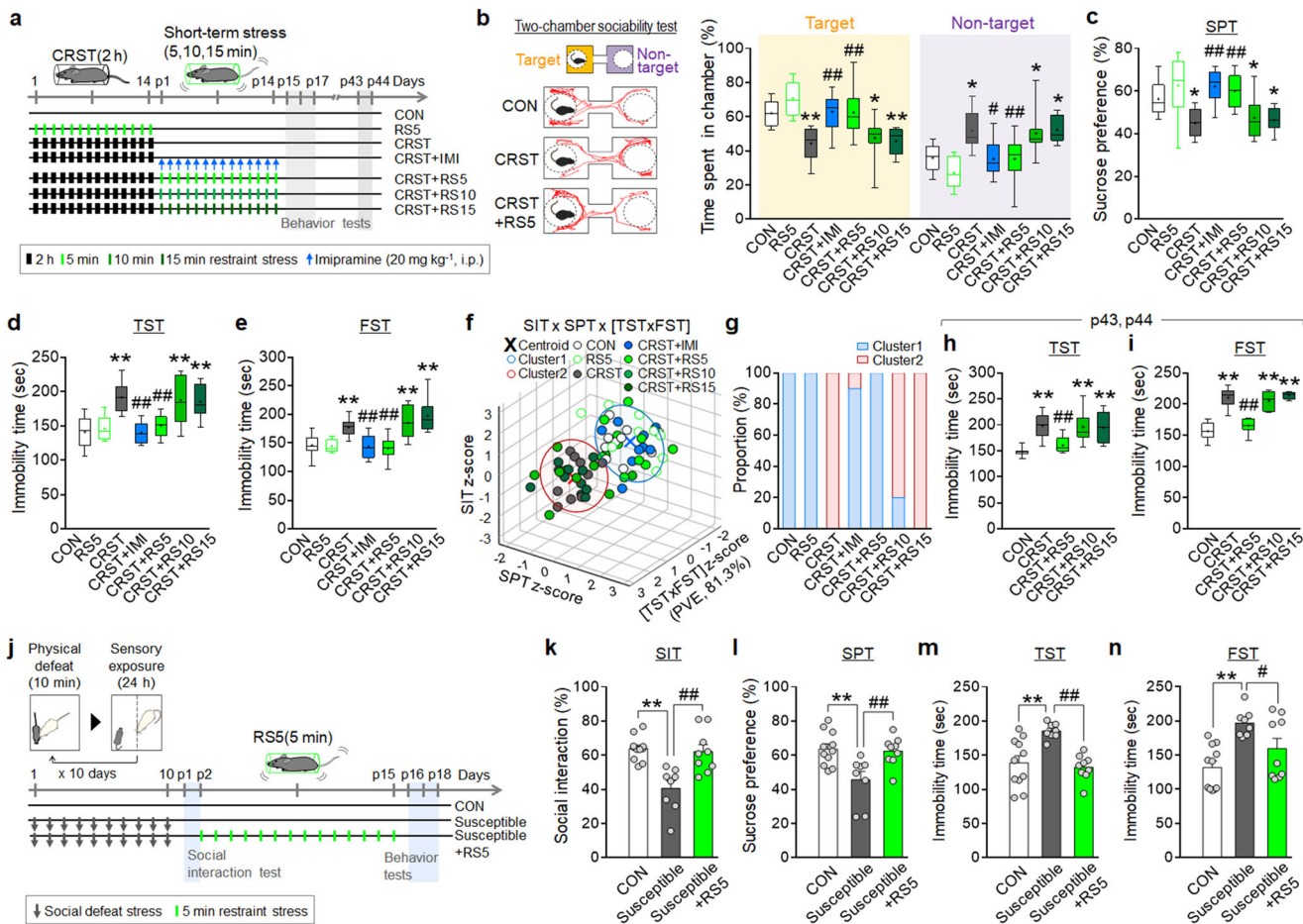

**Fig. 1 Repeated treatment with short-term behavioral stress produces anti-depressive effects in stress-induced models of depression. a** Experimental design. Mice were exposed to 5-min or 2-h restraint for 14 days as depicted. CRST mice were treated for 14 days with 5-min, 10-min, or 15-min restraint or imipramine (RS5, RS10, RS15, and IMI, respectively), and they were placed in the behavioral tests. **b**–**g** Representative tracks and the % time spent in the target and nontarget fields in the two-chamber sociability test (**b**), the % sucrose-preference in the SPT (**c**), and immobility time in the TST (**d**) and FST (**e**) for the indicated groups on post-stress days 15–17. K-means clustering ($k = 2$) of individuals in the SIT × SPT × [TST × FST] matrix (**f**) and % composition of each group in the clusters (**g**). PCA was used for dimension reduction of the TST × FST components (PVE, 81.3%) ($n = 9$–10 animals per group). **h**–**I** Immobility time in the TST (**h**) and FST (**i**) for the indicated groups on post-stress days 43 and 44 ($n = 8$–10 animals per group). **j** Experimental design. Mice subjected to CSDS were placed in the social interaction test on day 11 and susceptible and resilient groups were selected on the basis of the social interaction ratio. The susceptible mice were treated with RS5 and then were placed in the behavioral tests. **k**–**n** The % time spent in the target chamber in the two-chamber SIT (**k**), the % sucrose-preference in the SPT (**l**), and immobility time in the TST (**m**) and FST (**n**) for the indicated groups ($n = 8$–11 animals per group). Data were mean ± SEM. Gray circles represent individual data points. *, the difference compared to control; #, the difference compared to CRST. *, #, $p < 0.05$; **, ##, $p < 0.01$ (One-way ANOVA followed by Newman–Keuls post hoc test). See Supplementary Data 4 for statistical details.

**Low-dose glucocorticoids mimicked RS5 effects**. Next, we investigated whether RS5 can modulate stress-induced changes of the HPA axis. The serum corticosterone (CORT) levels induced by 5-min or 15-min restraint in normal mice were much lower than those induced by 2-h restraint, but they cleared as slowly as the levels induced by 2-h restraint (Fig. 2a). The CORT level induced by a single 5-min restraint (S5) in CRST mice peaked at 124.2 ng per ml (ng ml$^{-1}$) after 10 min, which was higher than that (97.4 ng ml$^{-1}$) induced by S5 in normal mice. Moreover, the CORT level induced by S5 in CRST mice cleared more slowly than that induced by S5 in normal mice (Fig. 2b, c), but RS5 treatment in CRST mice reverted the peak CORT level and its clearance time course to those of the control (Fig. 2c, d). These results suggest that RS5 rectifies the stress-induced dysregulation of the HPA axis.

CRST mice had increased basal serum CORT levels compared with naïve controls, and RS5 treatment in CRST mice restored the increased basal CORT levels to the controls (Fig. 2e). Interestingly, RS10 and RS15 treatment in CRST mice also significantly

suppressed the basal CORT levels, although their effects were less dramatic than those of RS5 (Fig. 2e). CRST mice showed increased expression of corticotropin-releasing hormone (CRH) and arginine vasopressin (AVP) in the paraventricular nucleus (PVN), and RS5 treatment in CRST mice suppressed their expression to the levels found in naïve controls. RS10 and RS15 treatment partially reduced CRH and AVP expression (Fig. 2f).

We tested whether post-stress treatment with low-dose CORT would mimic the physiological and behavioral effects of RS5. Intraperitoneal injection of CORT in naïve mice increased the serum CORT level in a dose-dependent manner. The serum CORT level following injection of CORT at 0.1 mg per kg (mg kg$^{-1}$) of body weight was close to that induced by 5-min restraint (Fig. 2a, g). Repeated injection of CRST mice with 0.1 mg kg$^{-1}$ of CORT, or even 0.5 or 0.1 mg kg$^{-1}$ of CORT, produced anti-depressive effects in the SIT, SPT, TST, and FST, and those behavioral rescues were comparable to those achieved by RS5 (Fig. 2h–l). Repeated injection of low-dose CORT in CRST mice also restored their increased basal CORT and reduced the increased weight of their adrenal glands to

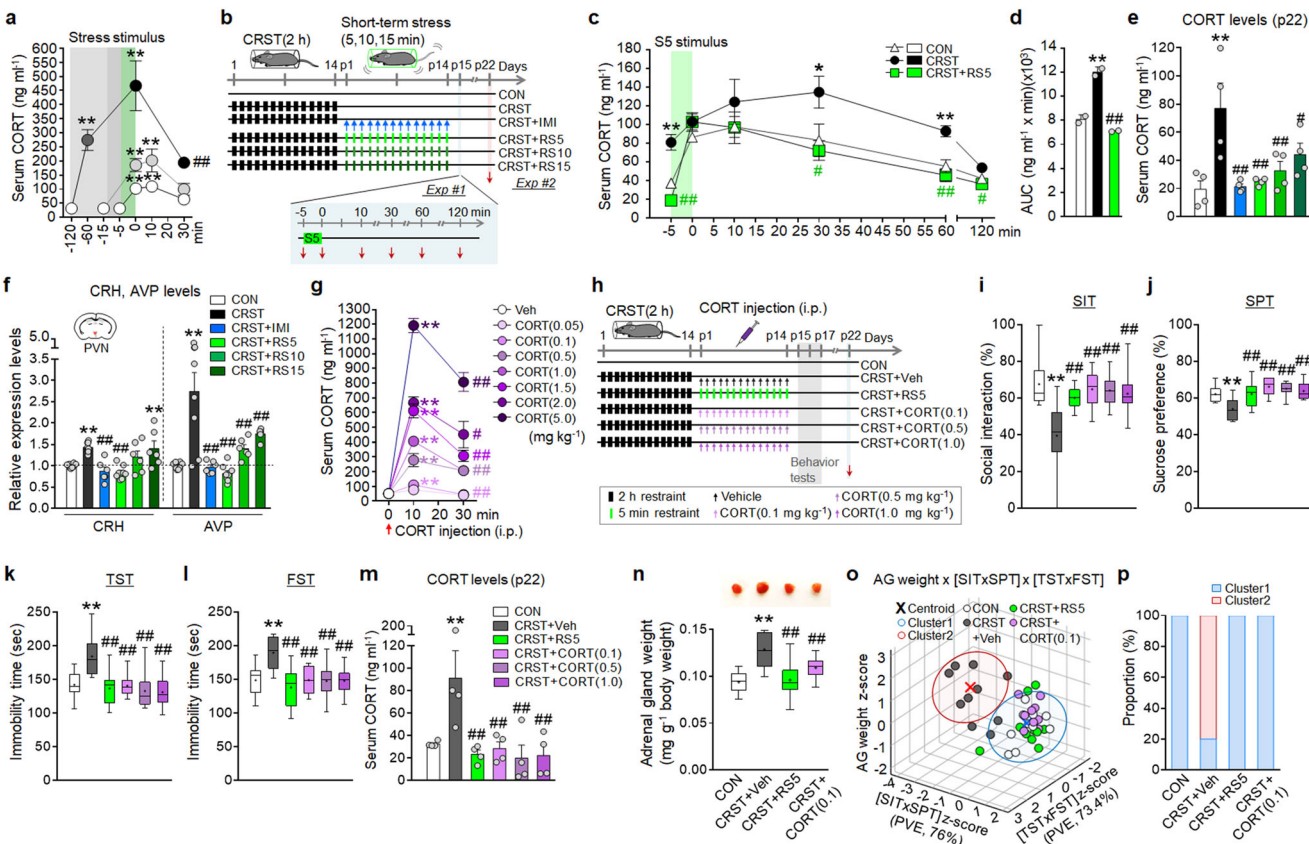

**Fig. 2 RS5 or low-dose CORT treatment normalizes stress-induced dysregulation of the HPA axis. a** Serum CORT levels in mice treated with 5-min, 15-min, 60-min, and 120-min restraint ($n = 8$–9 animals per group). **b**–**f** Experimental design (**b**). Time course of serum CORT levels in the indicated groups after exposure to a single 5-min restraint (S5) (Exp #1) (**c**), and the area under the curve (AUC) between −5 min and 120 min (**d**) ($n = 7$–8 animals per group). Basal serum CORT levels (**e**), and transcripts levels of CRH and AVP in the PVN (**f**) for the indicated groups (Exp #2) ($n = 7$–8 animals per group). **g** Changes in serum CORT levels in mice after injection with CORT injection (0.05, 0.1, 0.5, 1.0, 1.5, 2.0, 5.0 mg kg$^{-1}$, i.p.) ($n = 8$–9 animals per group). **h**–**p** Experimental design (**h**). CORT was injected (0.1, 0.5, 1.0 mg kg$^{-1}$ per day, i.p.) for 14 days in CRST mice. The % time of social interaction in the SIT (**i**), % sucrose-preference in the SPT (**j**), and immobility time in the TST (**k**) and FST (**l**) for the indicated groups ($n = 10$ animals per group). Basal serum CORT levels ($n = 8$ animals per group) (**m**) and adrenal gland (AG) weight ($n = 10$ animals per group) (**n**) of the indicated groups on post-stress day 22. K-means clustering of individuals in the [AG weight] x [SIT × SPT] × [TST × FST] matrix (**o**) and % composition of each group in the clusters (**p**). PCA was used for dimensional reduction of the SIT and SPT components (PVE; 76.0%) and the TST and FST components (PVE; 73.4%). Data were mean ± SEM. Gray circles represent individual data points. *, the difference compared to control; #, the difference compared to CRST. *, #, $p < 0.05$; **, ##, $p < 0.01$ (One-way ANOVA followed by Newman–Keuls post hoc test). See Supplementary Data 4 for statistical details.

those of naïve controls (Fig. 2m, n). PCA and K-means clustering of behavioral and physiological factors in the [SIT x SPT] x [TST x FST] x [adrenal gland weight] matrix indicated that behavioral recovery with CORT treatment proceeds with the restoration of adrenal gland weight at the individual level (Fig. 2o, p).

**RS5 effects required activation of the HPA axis.** NBI27914 (an inhibitor of CRH receptor 1) or RU486 (an inhibitor of GR) treatment in CRST mice unexpectedly heightened the increased basal CORT level, but it produced partial suppression of stress-induced depressive-like phenotypes in the SIT, SPT, TST, and FST. Administration of NBI27914 or RU486 during RS5 treatment in CRST mice suppressed the remediating effects of RS5 on the basal CORT levels and dissipated the anti-depressive effects of RS5 in the behavioral tests (Supplementary Fig. 4a–h).

Mice subjected to CRST followed by adrenalectomy (ADX) surgery had a low basal CORT level and showed no CORT release after 5-min restraint (Fig. 3a, b). RS5 treatment in CRST mice that received ADX did not rescue depressive-like phenotypes in the SIT, SPT, TST, and FST (Fig. 3c–h). ADX surgery alone in CRST mice did not improve depressive-like behavior (Fig. 3c–h).

PCA and K-means clustering indicated that CRST-treated individuals that received ADX were grouped into the cluster containing CRST mice. These results suggest that HPA axis activation is necessary for RS5 to produce therapeutic effects, whereas simple CORT depletion by ADX is ineffective.

**RS5 treatment activated core parts of the limbic system.** Since RS5 and RS15 produced differential effects on stress-induced depressive-like behavior (Fig. 1a–i), we tested whether these two different short-term stress paradigms could be used to identify specific brain regions mediating RS5 effects. To address this, we used a stimulus-induced c-Fos mapping strategy. Repeated treatment with 5-min restraint or 15-min restraint in CRST mice increased c-Fos expression in the prelimbic cortex (PL), infra-limbic cortex (IL), dentate gyrus (DG), ventral subiculum (vSub), dorsal bed nucleus of the stria terminalis (dBNST), ventral BNST (vBNST), PVN, and other parts of the limbic system (Fig. 4a–c and Supplementary Data 1). Interestingly, the c-Fos levels induced by repeated 5-min restraint in the PL, vSub, dBNST, and PVN were higher than those induced by repeated 15-min restraint or a single 5-min or 15-min restraint.

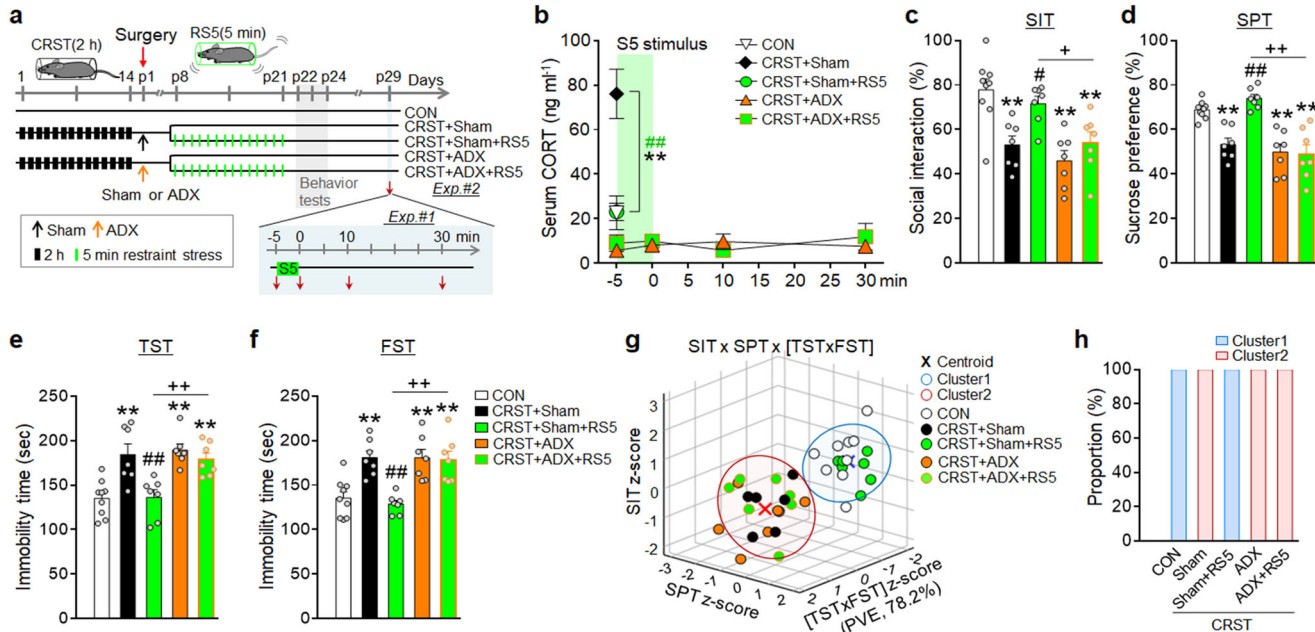

**Fig. 3 Adrenalectomy (ADX) dissipates the anti-depressive effects of RS5 in CRST mice. a** Experimental design. Mice were exposed to CRST, followed by ADX or sham surgery, and RS5 treatment, and then were placed in the behavioral tests. Red arrows (↓) in Exp #1, Serum collection points. Mice were sacrificed after exposure to a single 5-min restraint (S5). **b–h** The basal CORT levels at −5 min, and the time course of serum CORT levels in the indicated groups after exposure to S5 (Exp #1) (**b**) (n = 7–10 animals per group). The % time of social interaction in the SIT (**c**), the % sucrose-preference in the SPT (**d**), and immobility time in the TST (**e**) and FST (**f**) for the indicated groups (Exp #2). K-means clustering of individuals in the SIT × SPT × [TST × FST] matrix (**g**) and % composition of each group in the clusters (**h**). The TST and FST components were transformed into linear eigenvectors using PCA (PVE; 78.2%) (n = 7–10 animals per group). Data were mean ± SEM. Gray circles represent individual data points. *, the difference compared to control; #, the difference compared to CRST; +, the difference compared to CRST + RS5. *, #, +, $p < 0.05$; **, ##, ++, $p < 0.01$ (One-way ANOVA followed by Newman–Keuls post hoc test). See Supplementary Data 4 for statistical details.

Repeated injection of low-dose CORT (0.1, 0.5, or 1.0 mg kg$^{-1}$) in CRST mice also increased c-Fos expression in the PL, IL, basolateral amygdala (BLA), nucleus accumbens core (NAcc), DG, vSub, dBNST, vBNST, and PVN (Supplementary Fig. 5a–c), the same regions activated by repeated 5-min restraint (Supplementary Data 1).

The PL, vSub, dBNST, and PVN are functionally interrelated in stress responses[10,11,28,29]. The PL is an upstream limbic structure that has the potential to regulate HPA axis responses to psychogenic stressors[10,28,30,31]. Therefore, we investigated which cell types of PL neurons responded to RS5. Among the c-Fos-positive neurons in the PL activated by repeated 5-min restraint, 74.4% were glutamatergic, and 17.6 % were GABAergic. Among the glutamate neurons, 46.8% were c-Fos-positive, whereas 44.2% of GABA neurons were c-Fos positive (Fig. 4d–g).

**RS5 treatment restored altered gene expression in the PL.** We investigated the molecular targets changed by CRST and RS5 in the PL. Microarray analysis revealed that CRST treatment up- or down-regulated 264 genes by ≥1.2-fold compared to the control (Supplementary Data 2). A heatmap presentation of these gene expression profiles with the unbiased alignment of the respective genes whose expression was changed by RS5 in CRST mice highlighted that the stress-induced up- and down-regulated 264 genes by ≥1.2-fold underwent a cross reversal after RS5 treatment (Fig. 5a, b). In a reverse approach, RS5 treatment in CRST mice up- or downregulated 722 genes by ≥1.2-fold (Supplementary Data 3). A follow-up heatmap presentation of those gene expression profiles with the unsupervised alignment of respective genes whose expression was changed by CRST indicated that RS5-induced up- and down-regulated genes include those changed by CRST (Fig. 5a, c).

Gene Ontology (GO) enrichment analysis combined with K-means clustering using the STRING database[32] showed that the 264 and 722 genes described above could be grouped into multiple clusters presenting various protein–protein interaction (PPI) networks. A serial K-means clustering indicated that grouping those genes into 5–10 clusters (k = 5–10) built up the PPI networks containing functional modules relevant to the assumption of stress or glucocorticoid-related responses (Supplementary Fig. 6a). In the classification with 8 clusters (k = 8), clusters 5 and 7, which contained 28 and 25 members, respectively, carried functional modules with the GO terms "response to stress" or "response to glucocorticoid" (Fig. 5d, e). The remaining clusters are summarized in Supplementary Fig. 6b, c. Using the STRING database, the functional PPI networks formed by clusters 5 and 7 were enriched with an additional 16 proteins that could participate as nodes (Fig. 5f).

**RS5 treatment upregulated GR expression in the PL.** Chronic stress decreases GR expression and increases FK506-binding protein 5 (Fkbp5), a cellular factor that suppresses GR nuclear translocation[33], in the mPFC[34]. Of the genes for stress or glucocorticoid-related responses (Fig. 5f), GR expression in the PL was downregulated by CRST, whereas its expression was reversed to the control after RS5 or CORT (0.1 mg kg$^{-1}$) treatment. Fkbp5 and Fkbp4 expressions were upregulated and downregulated, respectively, by CRST, whereas their altered expression was restored after RS5 or CORT (0.1 mg kg$^{-1}$) treatment (Fig. 5g), suggesting that GR, Fkbp5, and Fkbp4 are critical players in stress coping responses to CRST and RS5. Transcript levels of ERK1 and ERK2 declined after CRST, and their expression was reversed by CORT (0.1 mg kg$^{-1}$) treatment, but not by RS5. The transcript levels of mineralocorticoid receptor

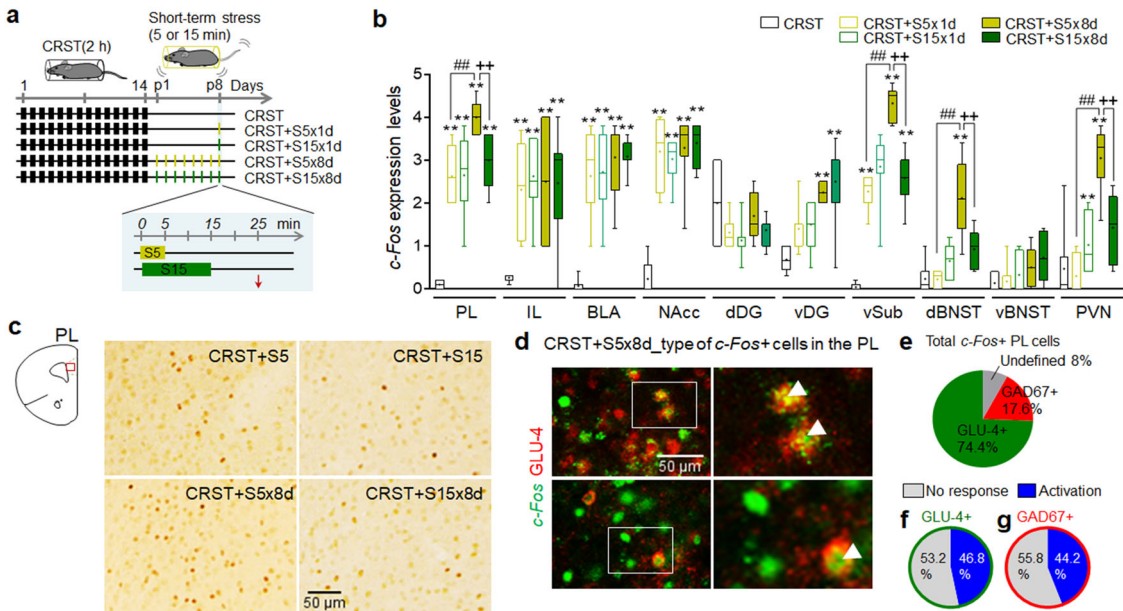

**Fig. 4 Repeated treatment with short-term behavioral stress recruits the brain regions regulating stress coping in CRST mice. a** Experimental design. CRST mice were treated with a single 5-min or 15-min restraint (S5 × 1d and S15 × 1d, respectively) on post-stress day 8, and then sacrificed 20 min and 10 min later, respectively. Another group of CRST mice was treated with 5-min or 15-min restraint for 8 days (S5 × 8d and S15 × 8d, respectively), and then sacrificed, respectively, 20 min and 10 min after the last restraint. The red arrow (↓), sample prep point. **b, c** Quantification of c-Fos expression levels in the PL, IL, BLA, NAcc, dorsal dentate gyrus (dDG), ventral dentate gyrus (vDG), vSub, dBNST, vBNST, and PVN (**b**). Photomicrographs showing c-Fos expression in the PL (red box) (C) of the S5 × 1d, S15 × 1d, S5 × 8d, and S15 × 8d groups (n = 4–6 animals per group). **d–g** Photomicrographs showing c-Fos induction in GLU-4-positive or GAD67-positive neurons in the PL (**d**), and their quantification levels (**e–g**) (n = 6 animals for a group). The details for c-Fos mapping are shown in Supplementary Data 1. Data were mean ± SEM. Gray circles represent individual data points. *, **, the difference compared to control; #, the difference compared to CRST; +, the difference between indicated groups. **, ##, ++, p < 0.01 (one-way ANOVA followed by Newman–Keuls post hoc test). See Supplementary Data 4 for statistical details.

(MR), heat shock protein 90α a1 (Hsp90aa1), heat shock protein 90α b1(Hsp90ab1), dual-specificity phosphatase 1 (Dusp1), and CaMKIIα tended to be changed by CRST, RS5, or CORT (0.1 mg kg⁻¹), although their changes overall were subtle (Fig. 5g, h).

Immunohistochemical analysis indicated that CRST decreased GR nuclear distribution and increased Fkbp5 expression, whereas RS5 and CORT (0.1 mg kg⁻¹) treatment reverted their altered expression to the controls (Fig. 5i–l). K-means clustering of GR and Fkbp5 expression indicated that individual cells of the CRST group were mostly distributed in the cluster presenting low GR and high Fkbp5 expression, whereas individual cells of the CON, RS5, or CORT groups were mostly distributed in the cluster presenting high GR and low Fkbp5 expression (Fig. 5m, n). Small interfering RNA (siRNA)-mediated knockdown of GR in the PL increased Fkbp5 expression (Fig. 5o–r). Furthermore, siRNA-mediated GR-knockdown in the PL increased CRH and AVP expression in the PVN (Fig. 5s), and produced increased immobility in the TST and FST (Fig. 5t, u). Together, these results support the notion that the reduction of GR expression in the PL promotes depressive-like behavior, and that CRST decreases GR expression, which in turn increases Fkbp5 expression, in PL neurons, whereas RS5 and low-dose CORT treatment restore the altered GR and Fkbp5 expression.

**RS5 reversed the stress-induced increase of p-CaMKIIα in the PL.** CaMKII and ERK1/2 are important signaling factors mediating stress coping responses in the mPFC[35,36]. Western blot analyses indicated that CRST treatment increased p-CaMKIIα and decreased p-ERK1/2 levels in the PL, whereas RS5 treatment restored the altered expression of p-CaMKIIα, but not p-ERK1/2, to naïve controls (Fig. 6a–c). Immunohistochemical analysis showed that p-CaMKIIα and p-ERK1/2 were expressed mostly in

glutamatergic neurons in the PL (Fig. 6d, e and Supplementary Fig. 7a, b). CRST increased p-CaMKIIα levels while decreasing GR expression in PL neurons, whereas the altered expressions of those factors were restored to the control levels after RS5 or CORT (0.1 mg kg⁻¹) treatment (Fig. 6f–i). K-means clustering followed by regression analysis indicated that GR expression in PL neurons was positively correlated with p-CaMKIIα levels at the single-cell level in the CON, CRST + RS5, and CRST + CORT groups, whereas in the CRST group, GR expression remained relatively low even at high p-CaMKIIα levels (Fig. 6j, k). siRNA-mediated knockdown of CaMKIIα in the PL of CRST mice restored the reduced GR expression, whereas siRNA-mediated knockdown of ERK1 or ERK2 did not change the reduced GR expression, suggesting that CaMKIIα, but not ERK1 and ERK2, is a critical player for stress-induced decreased GR expression in the PL. However, Fkbp5 expression was not changed by the knockdown of CaMKIIα nor ERK1, although its expression was partially affected by the suppression of ERK2 (Fig. 6l–q). Moreover, siRNA-mediated knockdown of CaMKIIα, but not ERK1 and ERK2, in the PL of CRST mice restored the reduced sociability and increased immobility in the TST and FST (Fig. 6r–t).

GABAergic interneurons in the prefrontal cortex directly regulate the neuronal activity of glutamatergic neurons in chronic stress and depression[37]. Since repeated 5-min restraint induced c-Fos expression in both glutamatergic and GABAergic neurons in the PL (Fig. 4d–g), we investigated whether PL GABAergic neurons have a role in regulating CRST and RS5 effects. CRST treatment decreased the expression of the GABAA receptor subunits GABARα1 and GABARβ2 in the PL, and RS5 or CORT (0.1 mg kg⁻¹) treatment restored their reduced expression (Fig. 7a).

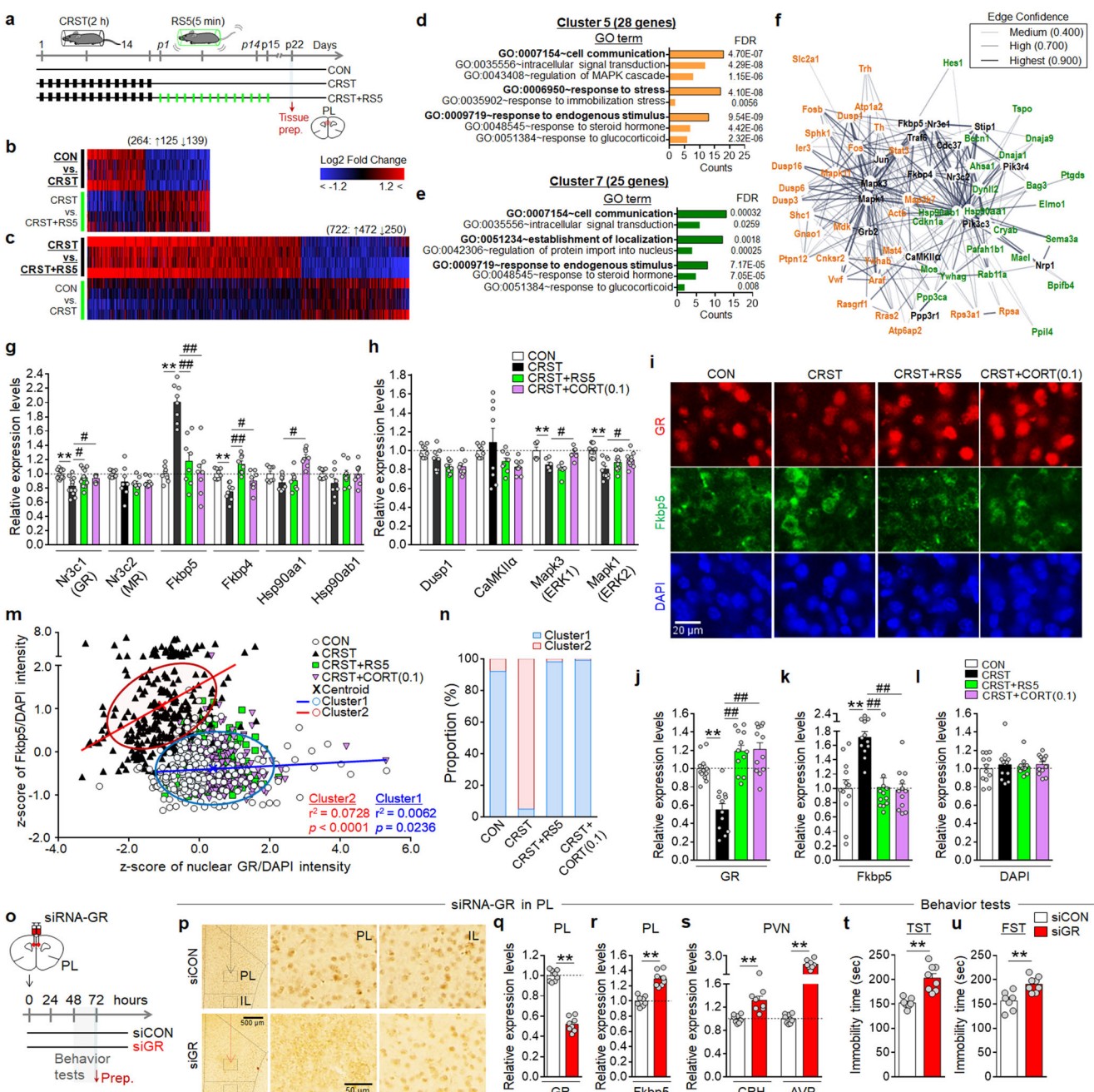

**Fig. 5 RS5 treatment reverses stress-induced gene expression changes in the PL. a–c** Experimental design (**a**). Heatmap showing the expression profiles of genes changed by ≥1.2-fold after CRST (**b**) and those changed by RS5 treatment in CRST mice (**c**), and follow-up unbiased alignment with expression levels of the respective genes changed by RS5 (**b**) and CRST (**c**), respectively. ↑ and ↓, indicate up- and down-regulation, respectively. **d–f** Two clusters identified from GO enrichment analysis with weighted *K*-means clustering of gene expression profiles (**d**, **e**). The PPI networks formed by the genes of clusters 5 (orange, 28 genes) and 7 (green, 24 genes) and 16 genes (black) were identified from an extended search of the STRING database at a confidence level of >0.700 (**f**). **g, h** Transcript levels of GR(Nr3c1), MR(Nr3c2), Fkbp5, Fkbp4, Hsp90aa1, and Hsp90ab1 (**g**) and Dusp1, CaMKIIα, ERK1(Mapk3), and ERK2(Mapk1) (**h**) in the PL of the indicated groups (*n* = 8–12 per group). **i–n** Immunofluorescence images showing total GR (red) and Fkbp5 (green) expression in the PL for the indicated groups (**i**). DAPI, blue. Quantification levels of GR (**j**), Fkbp5 (**k**), and DAPI intensity (**l**) (*n* = 4–6 animals per group). Two distinct clusters with differential expression of GR and Fkbp5 across individual cells for the indicated groups (**m**), and the % composition of individual cells of each group in the clusters (**n**) (*n* = 248–309 cells from 4–6 animals per group). **o–u** Experimental design (**o**). Immunohistochemical images showing siRNA-mediated knockdown of GR expression (**p**) and GR transcript levels (**q**) in the PL. Transcript levels of Fkbp5 in the PL (**r**), and of CRH and AVP in the PVN (**s**) of mice injected with siRNA-GR in the PL. Immobility time in the TST (**t**) and FST (**u**) for the indicated groups (*n* = 6–8 per group). Data were mean ± SEM. *, the difference compared to control; #, the difference compared to CRST. *, #, *p* < 0.05; **, ##, *p* < 0.01 (Two-sided Student's *t*-test; one-way ANOVA followed by Newman–Keuls post hoc test). See Supplementary Data 4 for statistical details.

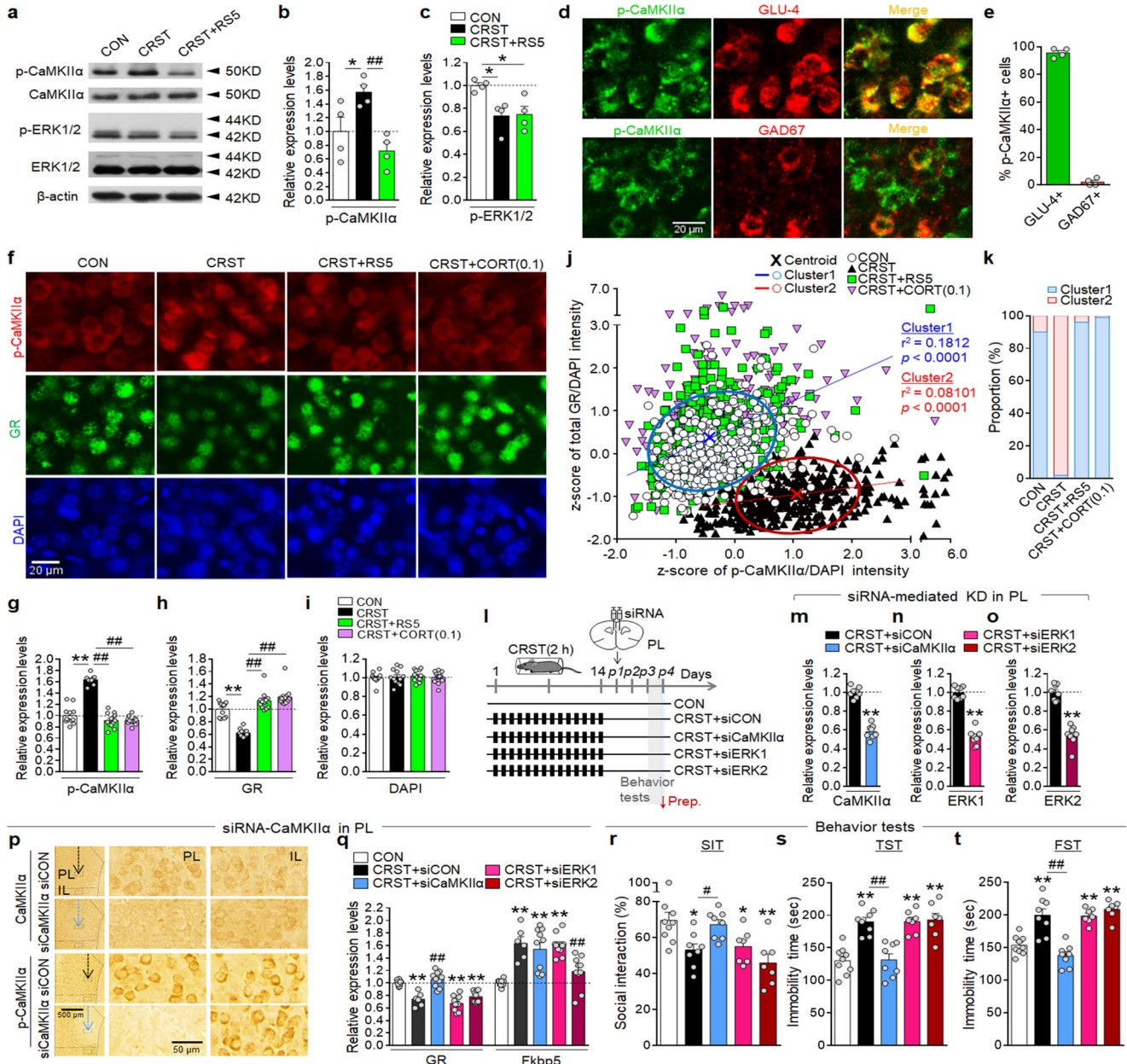

**Fig. 6 RS5 restores increased p-CaMKIIα expression in PL neurons. a–c** Western blots showing expression levels of p-CaMKIIα, CaMKIIα, p-ERK1/2, ERK1/2, and β-actin in the PL of the indicated groups (**a**). Quantification levels of p-CaMKIIα (**b**) and p-ERK1/2 expression (**c**). Western blots were representative of four technical repeats carried out using two sets of the indicated groups, each set with tissue samples from three pooled animals per group. β-actin was included as an internal loading control. **d, e** Immunofluorescence staining of p-CaMKIIα (green) in PL neurons stained with GLU-4 (a marker of glutamate neurons; red) or GAD67 (red) (**d**) and their quantification levels (**e**) ($n = 4$ per group). **f–k** Immunofluorescence images showing p-CaMKIIα (red) and total GR (green) expression in PL neurons of the CON, CRST, RS5, and CORT (0.1 mg kg$^{-1}$) groups (**f**). DAPI, blue. Quantification levels of p-CaMKIIα (**g**), GR (**h**), and DAPI intensity (**i**) ($n = 4–6$ animals per group). Two distinct clusters with differential expression of p-CaMKIIα and total GR across individual cells of the indicated groups (**j**) and the % composition of individual cells of each group in the clusters (**k**) ($n = 345–390$ cells from 4–6 animals per group). **l–t** Experimental design (**l**). Transcript levels showing siRNA-mediated knockdown of CaMKIIα (**m**), ERK1 (**n**), and ERK2 (**o**) in the PL of CRST-treated mice. Immunohistochemical images showing siRNA-mediated knockdown of CaMKIIα and p-CaMKIIα in siRNA injection sites (**p**). Transcript levels of GR and Fkbp5 (**q**) in the PL for the indicated groups ($n = 4–8$ animals per group, 3–5 repeats). The % time of social interaction in the SIT (**r**), and immobility time in the (**s**) and FST (**t**) for the indicated groups ($n = 7–9$ animals per group). Data were mean ± SEM. Gray circles represent individual data points. *, the difference compared to control; #, the difference compared to CRST. *, #, $p < 0.05$, **, ##, $p < 0.01$ (Two-sided Student's $t$-test; one-way ANOVA followed by Newman–Keuls post hoc test). See Supplementary Data 4 for statistical details.

We tested the role of GABA receptors in the production of RS5 effects. Infusion of picrotoxin (an inhibitor of GABAA receptor) into the PL through a pre-implanted cannula in mice that had been subjected to CRST followed by RS5 treatment increased p-CaMKIIα levels, but not p-ERK1/2, in the PL compared to those in the CRST plus RS5-treated group (Fig. 7b–e). Furthermore, the picrotoxin infusion in the PL of CRST mice abolished the anti-depressive effects of RS5 (Fig. 7f, g). These results

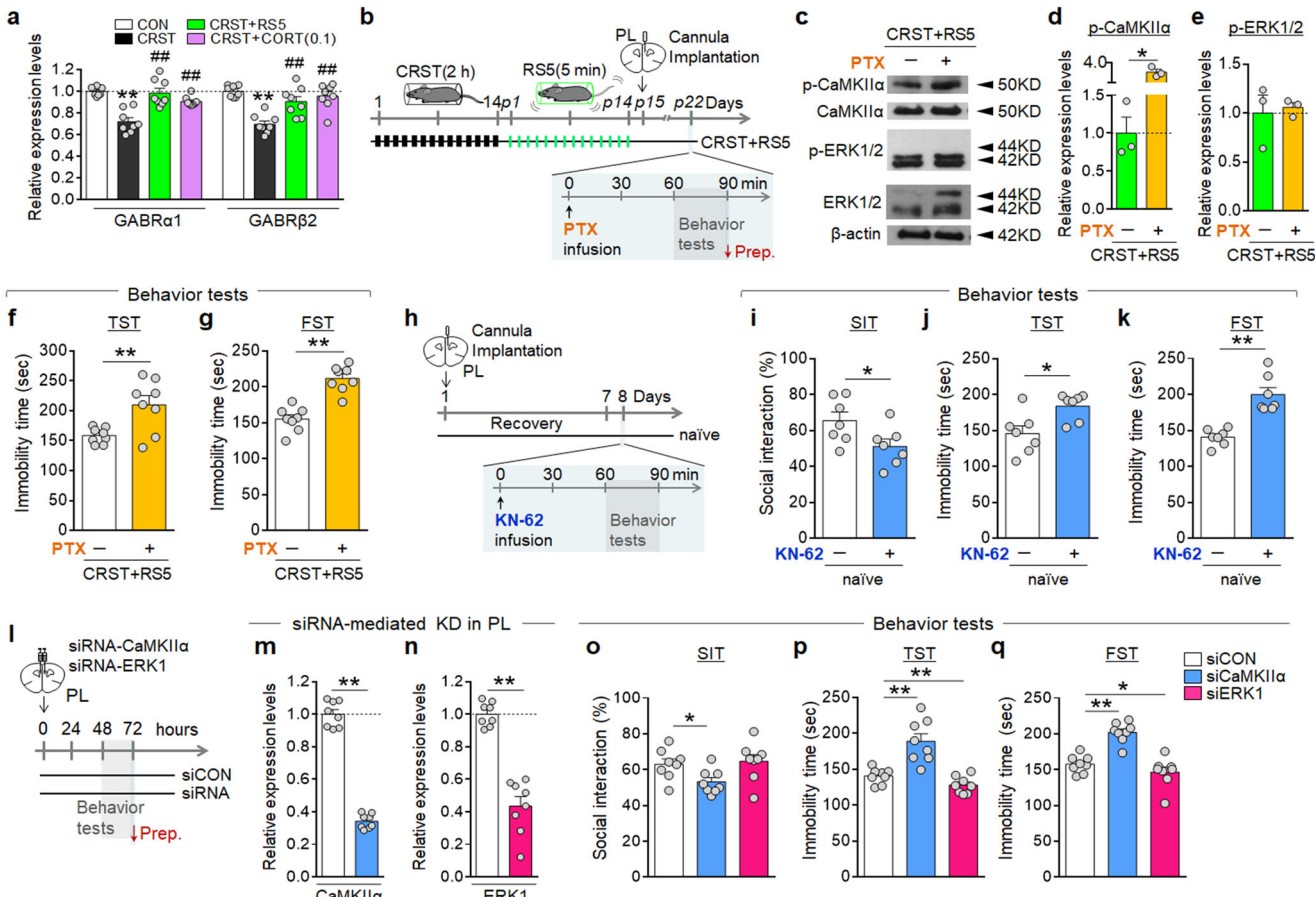

**Fig. 7 CaMKIIα in the PL is a critical player in regulating depressive behavior. a** Transcript levels of GABRα1 and GABRβ2 in the PL of the CON, CRST, RS5, and CORT (0.1 mg kg⁻¹) groups. Tissue sample groups were prepared from the experiments depicted in Fig. 4g, h ($n = 7$–12 per group). **b–g** Experimental design (**b**). Picrotoxin (PTX) was infused through a pre-implanted cannula into the PL of CRST and RS5-treated mice, and behavior tests were carried out 60–90 min after drug infusion on post-stress day 22 (**b**). Western blots showing the expression levels of p-CaMKIIα, CaMKIIα, p-ERK1/2, ERK1/2, and β-actin in the PL of the indicated groups (**c**). Quantification levels of p-CaMKIIα (**d**) and p-ERK1/2 (**e**). Western blots were representative of three technical repeats carried out using tissue samples from four pooled animals per group. β-actin was included as an internal loading control. Immobility time in the TST (**f**) and FST (**g**) for the indicated groups ($n = 8$ animals per group). **h–k** Experimental design (**h**). KN62 (2.5 nmol per injection) was infused into the PL through a pre-implanted cannula. After 60 min, behavioral tests were performed in the order SIT, TST, and FST. The % time of social interaction in the SIT (**i**) and immobility time in the TST (**j**) and FST (**k**) for the indicated groups ($n = 7$ animals per group). **l–q** Experimental design (**l**). Transcript levels showing siRNA-mediated knockdown of CaMKIIα (**m**) and ERK1 (**n**) in the PL, and behavioral tests were carried out 2 days later. The % time of social interaction in the SIT (**o**), and immobility time in the TST (**p**) and FST (**q**) for the indicated groups ($n = 8$ animals per group). Data were mean ± SEM. Gray circles represent individual data points. *, the difference compared to control; #, the difference compared to CRST. *, #, $p < 0.05$, **, ##, $p < 0.01$ (Two-sided Student's t-test; one-way ANOVA followed by Newman–Keuls post hoc test). See Supplementary Data 4 for statistical details.

support that p-CaMKIIα upregulation in PL neurons by CRST can be induced by decreased GABAA receptor function and RS5 restores p-CaMKIIα levels by increasing GABAA receptor function in the PL.

Next, we investigated the role of CaMKIIα in PL neurons in regulating depressive behavior. Local infusion of KN62 (an inhibitor of CaMKII) into the PL through a pre-implanted cannula in normal mice reduced social interaction in the SIT, and increased immobility in the TST and FST (Fig. 7h–k). Consistently, siRNA-mediated knockdown of CaMKIIα, but not ERK1, in the PL produced depressive-like behavior in the SIT, TST, and FST (Fig. 7l–q). These results suggest that CaMKIIα in PL neurons is a critical player for affective behavior.

Repeated stress changes the expression of GluN glutamate receptors in the prefrontal cortex[21]. Therefore, we investigated whether RS5 treatment could change the expression of GluN receptors in the PL. CRST treatment reduced the expression of the GluN subunits NR1, NR2A, and NR2B in the PL, whereas RS5 or CORT (0.1 mg kg⁻¹) treatment restored the reduced

expression of those GluN receptor subunits to the control levels (Supplementary Fig. 7c–j).

**Activation of PL neurons was required for RS5 effects.** Given that PL neurons distinctively responded to repeated 5-min restraint (Fig. 4) while changes in GR, FKBP5, CaMKIIα, GABAA, and GluN receptors in PL neurons were involved in RS5 effects (Figs. 5–7 and Supplementary Fig. 7c), we investigated whether RS5 effects were produced by activation of PL neurons. AAV8-CaMKIIα-hM3D(Gq) or AAV8-hSyn-hM3D(Gq) excitatory vector was transduced in the PL region. The mice were then subjected to CRST and treated with Clozapine N-oxide (CNO) as depicted. CNO injection in mice with the viral vector injection increased c-Fos expression in PL neurons (Supplementary Fig. 8a–f). CNO injection in mice with CaMKIIα promoter-driven hM3D(Gq) expression, but not the human synapsin (hSyn) promoter which is active in both glutamatergic and GABAergic neurons, produced increased sociability in the SIT and decreased immobility time in the TST and FST (Supplementary

Fig. 8g–k). Following CNO washout after 7 days, mice with the CaMKIIα-hM3D(Gq) expression showed a relapse of depressive behavior and had an increase in basal CORT levels relative to naïve controls (Supplementary Fig. 8l–q). These results suggest that stimulation of PL glutamatergic neurons in CRST mice can rescue depressive-like phenotypes.

AAV8-CaMKIIα-hM4D(Gi) inhibitory vector was transduced in the PL of mice. They were then subjected to CRST and treated with RS5 or RS5 with CNO as depicted (Fig. 8a, b). CNO injection in mice with hM4D(Gi) expression during RS5 treatment decreased the RS5-induced *c-Fos* expression in PL neurons (Fig. 8c, d). CNO-mediated inhibition of PL neurons in mice with the hM4D(Gi) expression during RS5 treatment blocked the RS5-induced suppression of the basal CORT level and dissipated the anti-depressive effects of RS5 in the SIT, SPT, TST, and FST (Fig. 8e–k). These results suggest that activation of PL neurons is necessary to produce anti-depressive effects of RS5.

Next, we investigated if CNO itself modifies stress-induced depressive phenotypes. Post-stress treatment with CNO at 3 mg kg$^{-1}$ per injection, the highest dose we used in the present study, for 14 days in CRST mice did not affect stress-induced depressive-like behavior (Supplementary Fig. 9), eliminating the possibility of nonspecific action of CNO or its metabolite effects.

**Multiple PL outputs mediated RS5 effects.** The PL sends collaterals to the dBNST, BLA, and NAcc (Supplementary Fig. 10a–g), the brain regions recruited by repeated treatment

with 5-min restraint (Fig. 4a, b). Therefore, we investigated whether these PL outputs regulate the physiological and behavioral effects of RS5.

The dBNST, which receives glutamatergic inputs from the PL (Supplementary Fig. 10a–d, h–l) contains GABA neurons that project to the PVN[10]. To test the role of the PL→dBNST circuit in RS5 effects, the AAV-DIO-hM4D(Gi)-mCherry inhibitory vector was transduced in PL→dBNST neurons using a retrograde WGA-Cre vector. The mice were then subjected to CRST followed by RS5 treatment (Fig. 9a–c). CNO injection in mice with the hM4D(Gi) expression in the PL→dBNST suppressed RS5-induced *c-Fos* expression in the dBNST, but not the PL and PVN (Fig. 9d–f), suggesting that the PL→dBNST circuit can be functionally inhibited by CNO injection. CNO-mediated inhibition of the PL→dBNST circuit during RS5 treatment in CRST mice blocked the RS5-induced recovery of the basal CORT level and the anti-depressive effects of RS5 in the SIT, SPT, TST, and FST (Fig. 9g–m).

Next, we tested whether stimulation of the PL→dBNST circuit in CRST mice mimicked the anti-depressive effects of RS5. AAV-DIO-hM3D(Gq)-mCherry excitatory vector was transduced in PL→dBNST neurons using a retrograde WGA-Cre vector. The mice were then subjected to CRST (Supplementary Fig. 11a–c). CNO injection in CRST mice with the hM3D(Gq) expression in the PL→dBNST increased *c-Fos* expression in the dBNST, but not the PL, NAcc, and BLA (Supplementary Fig. 11d–h). CNO-mediated stimulation of the PL→dBNST circuit in CRST mice

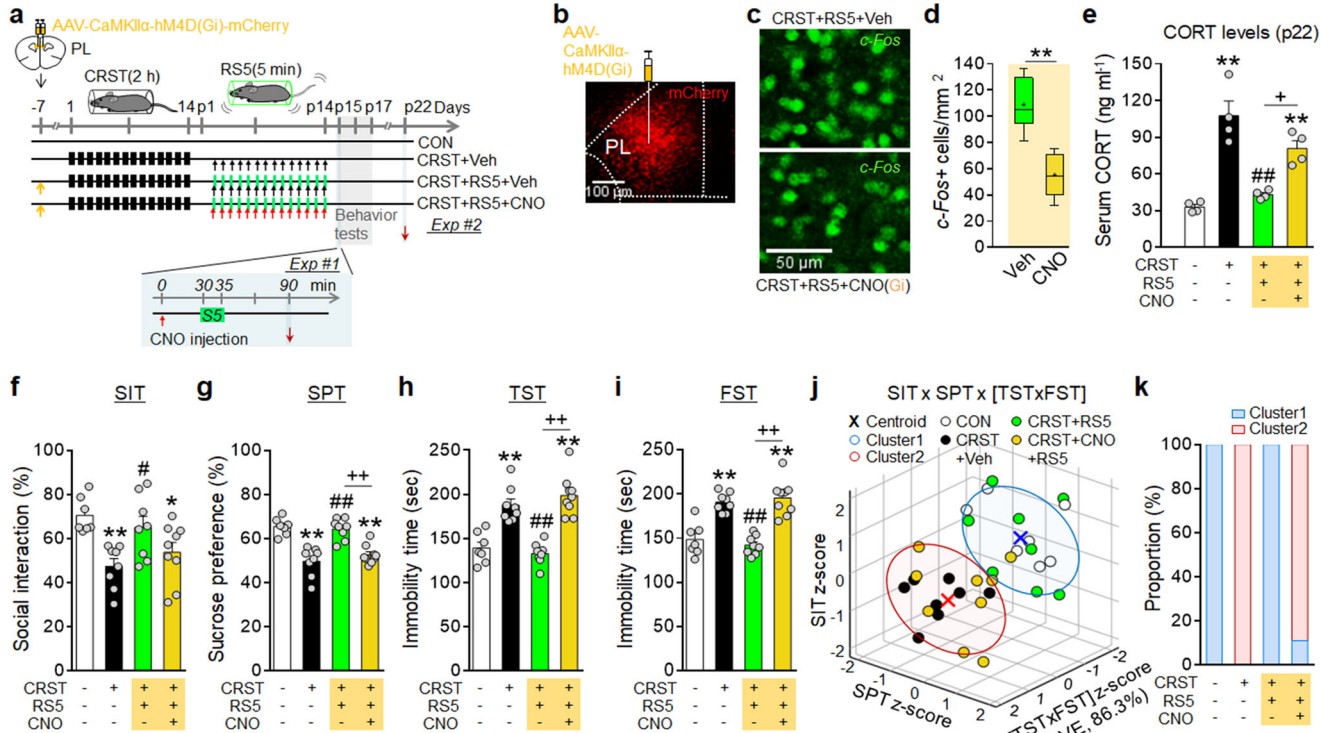

**Fig. 8 Chemogenetic inhibition of PL neurons blocks the anti-depressive effects of RS5. a** Experimental design. AAV8-CaMKIIα-hM4D(Gi)-mCherry inhibitory vector was injected into the PL. Mice were then subjected to CRST, followed by RS5 treatment with Veh or CNO injection. CNO, 3 mg kg$^{-1}$ per injection (i.p.). Red arrows (↓) in Exp #1, sample prep point and red arrow (↓) on post-stress day 22, serum collection point. **b–e** Photomicrograph showing mCherry expression by the injected vector in the PL (**b**). RS5-induced *c-Fos* expression levels in the PL after CNO or Veh injection (Exp #1) (**c, d**) (*n* = 4–6 per group). Basal serum CORT levels in the indicated groups on post-stress day 22 (Exp #2) (**e**) (*n* = 7–8 animals per group). **f–k** The % time of social interaction in the SIT (**f**), the % sucrose-preference in the SPT (**g**), and immobility time in the TST (**h**), and FST (**i**) for the indicated groups (Exp #2). K-means clustering of individuals in the SIT × SPT × [TST × FST] matrix (**j**) and % composition of each group in the clusters (**k**) (*n* = 7–9 animals per group). Data were mean ± SEM. Gray circles represent individual data points. *, the difference compared to control; #, the difference compared to CRST. *, #, *p* < 0.05, **, ##, *p* < 0.01 (Two-sided Student's *t*-test; one-way ANOVA followed by Newman–Keuls post hoc test). See Supplementary Data 4 for statistical details.

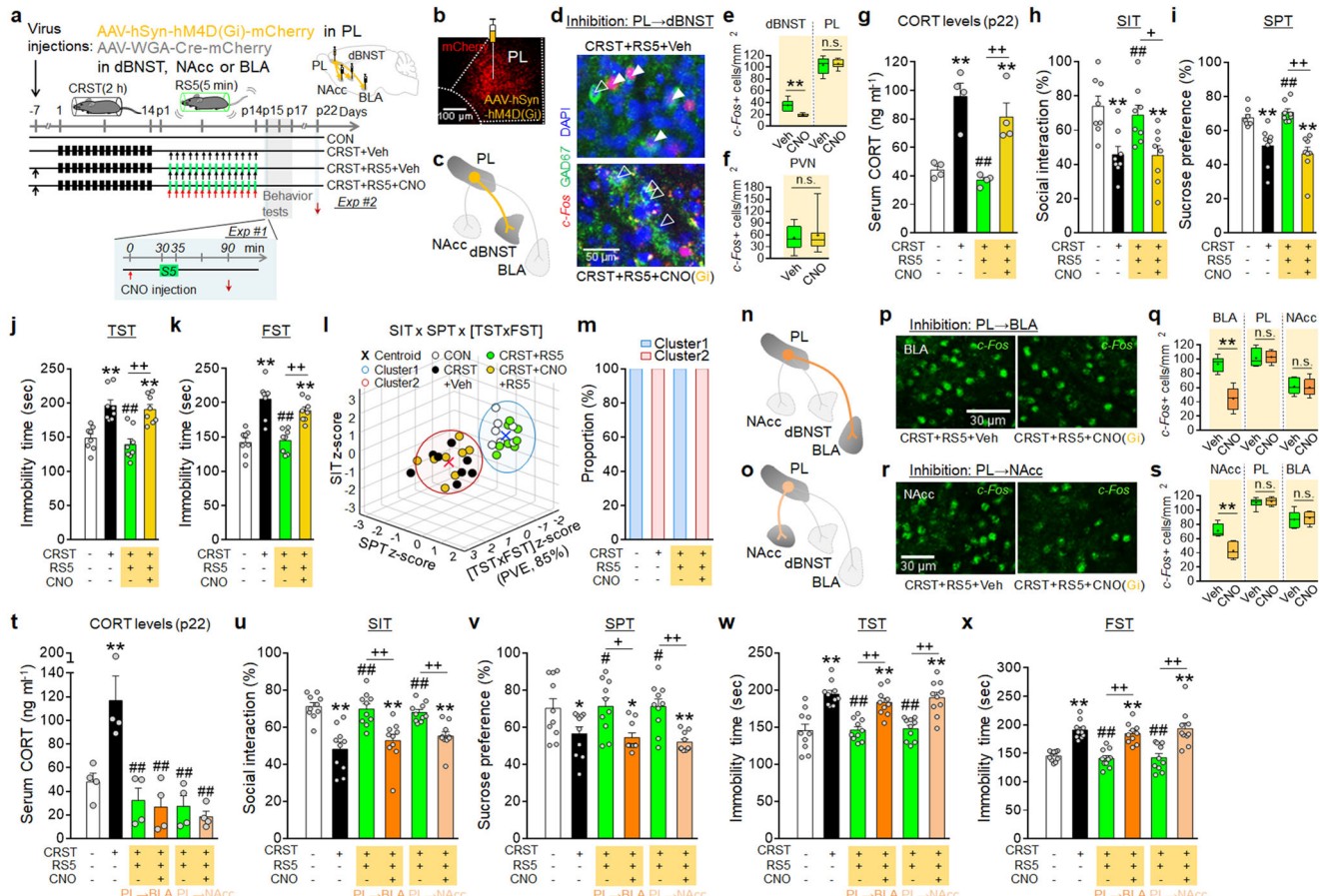

**Fig. 9 Chemogenetic inhibition of PL neurons projecting to the dBNST, BLA, or NAcc blocks the anti-depressive effects of RS5. a** Experimental design. AAV8-hSyn-hM4D(Gi)-mCherry injection into the PL and a retrograde Cre vector injection into the dBNST, BLA, or NAcc. Mice were then subjected to CRST, followed by RS5 treatment with Veh or CNO injection. **b–f** mCherry expression by the injected vector in the PL (**b**). Diagram for the PL→dBNST circuit (**c**) labeled with hM4D(Gi) expression. Immunofluorescence staining of *c-Fos* (red) and GAD67 (green) in the dBNST (**d**) of the indicated groups. Quantification of RS5-induced *c-Fos* expression in the dBNST, PL (**e**), and PVN (**f**) after CNO injection (Exp #1) (*n* = 4 per group). DAPI, blue. **g–m** Basal CORT levels (**g**), the % time of social interaction in the SIT (**h**), the % sucrose-preference in the SPT (**i**), and immobility time in the TST (**j**) and FST (**k**) for the indicated groups (Exp #2). *K*-means clustering of individuals in the SIT × SPT × [TST × FST] matrix (**l**) and the % composition of each group in the clusters (**m**) (*n* = 8 animals per group). **n–s** Diagrams for the PL→BLA (**n**) and PL→NAcc (**o**) circuits labeled with hM4D(Gi) expression. RS5-induced *c-Fos* expression in the BLA (**p**) and NAcc (**r**) after CNO or Veh injection for the PL→BLA (**p**) and PL→NAcc (**r**) groups. Quantification of RS5-induced *c-Fos* expression levels in BLA, PL, and NAcc (**q**) and NAcc, PL, and BLA (**s**) after CNO injection for the indicated groups (Exp #1) (*n* = 3-4 animals per group). **t–x** Basal CORT levels (**t**), the % time of social interaction in the SIT (**u**), the % sucrose-preference in the SPT (**v**), and immobility time in the TST (**w**) and FST (**x**) for the indicated groups (Exp #2) (*n* = 10 animals per group). Data were mean ± SEM. *, the difference compared to control; #, the difference compared to CRST. *, #, *p* < 0.05; **, ##, *p* < 0.01 (Two-sided Student's *t*-test; one-way ANOVA followed by Newman–Keuls post hoc test). n.s. not significant. See Supplementary Data 4 for statistical details.

produced increased sociability in the SIT and reduced immobility in the TST and FST (Supplementary Fig. 11i–m), thus abolishing depressive-like phenotypes of CRST mice. Together, these results suggest that activation of the PL→dBNST is required for the anti-depressive effects of RS5.

Next, we examined the roles of PL→BLA and PL→NAcc neurons in regulating RS5 effects. AAV-DIO-hM4D(Gi)-mCherry vector was transduced in the PL→BLA or PL→NAcc circuits using a retrograde WGA-Cre vector (Fig. 9a, n, o). The mice were then subjected to CRST followed by RS5 treatment as depicted (Fig. 9a). CNO injection in mice with the hM4D(Gi) expression in the PL→BLA decreased RS5-induced *c-Fos* expression in the BLA, but not the PL or NAcc (Fig. 9p–s). Similarly, CNO injection in mice with the hM4D(Gi) expression in the PL→NAcc decreased RS5-induced *c-Fos* expression in the NAcc, respectively, but not the PL and BLA, (Fig. 9p–s), suggesting that CNO injection functionally inhibits these PL circuits. CNO-mediated inhibition of the PL→BLA or PL→NAcc

circuit during RS5 treatment in CRST mice blocked the anti-depressive effects of RS5 in the SIT, SPT, TST, and FST, but the inhibition of the PL→BLA or PL→NAcc circuit during RS5 treatment did not change the RS5-induced recovery of basal CORT levels (Fig. 9t–x).

We tested whether the results obtained by the selective expression of DREADDs-hM4D(Gi) in a specific circuit using the retrograde WGA-Cre system can be validated by an optogenetic method. In mice subjected to CRST followed by RS5 treatment as depicted (Supplementary Fig. 12a), AAV-CaMKIIα-eNpHR-eYFP inhibitory vector or AAV-CaMKIIα-eYFP control was injected in the PL and an optic fiber was implanted in the NAcc (Supplementary Fig. 12). Mice with the CaMKIIα-eNpHR-eYFP expression exhibited an optic stimulation-dependent decrease of social interaction in the SIT, and optic stimulation-dependent decrease of mobility time in the TST and FST, whereas mice with the CaMKIIα-eYFP expression did not show those optic stimulation-dependent changes in the

behavioral tests (Supplementary Fig. 12). These results are consistent with the results of DREADD-hM4D(Gi)-mediated inhibition of the PL→NAcc circuit (Fig. 9u–x).

## Discussion

The finding that the behavioral method implementing short-term behavioral stress arousals and its resolutions can rectify pre-existing depressive-like phenotypes in multiple models of depression highlights the feasibility of fighting adaptively altered prior stress gains with behavioral stress. Chronic restraint produces depressive-like behavior that lasts for more than 3 months[38,39]. Despite such lasting adaptive changes by chronic stress, RS5, but not RS10 or RS15, rectified the stress-induced depressive-like behavior as did the antidepressant imipramine (Fig. 1). However, RS5 treatment did not produce anti-depressive effects when the HPA axis was blocked using adrenalectomy surgery or pharmacological means (Fig. 3 and Supplementary Fig. 4). Interestingly, repeated injection with low-dose GC (0.1 mg kg$^{-1}$) recapitulated the effects of RS5 (Fig. 2h–l). Paradoxically, however, repeated injection with GC at a dose higher than that induced by 10-min or 15-min restraint, and repeated injection with even 1.0 mg kg$^{-1}$ of GC, which was comparable to the GC level induced by 2-h restraint (Fig. 2a, g), also produced anti-depressive effects (Fig. 2h–l). Collectively, these results suggest that the behavioral method implementing short-term stress produces anti-depressive effects through a GC-dependent mechanism. However, the behavioral method activates other mechanisms beyond the GC-dependent changes.

Chronic stress produces various adaptive changes in the brain through the mechanism of GC effects[4–6]. Patients with depression have increased basal serum GC levels[40,41]. RU486 (mifepristone), a GR antagonist, is beneficial for patients with psychotic depression[42]. Consistently, chronic stress increased basal serum GC levels in mice (Fig. 2), whereas RU486 partially blocked stress-induced depressive-like behavior (Supplementary Fig. 4). Administration of high-dose GC in drinking water (35 μg ml$^{-1}$ per day) for 4 weeks[43] or subcutaneous injection of GC at a dose of 10, 20, or 40 mg kg$^{-1}$ per day for 21 days[44] in rats mirrors the stress-induced dysfunction of the HPA axis and produces depressive-like behavior. In line with this view, GC is regarded as a mediator of chronic stress[4–6,8]. In the present study, we demonstrated that repeated treatment with short-term behavioral stress or repeated injection with GC (0.1–1.0 mg kg$^{-1}$ per day) produced anti-depressive effects (Figs. 1 and 2h–l) and reversed stress-induced molecular changes in the PL (Fig. 5). The key findings are summarized in Fig. 10. These results raise the following important and related points. First, the fact that GC induction by behavioral stress and exogenous GC resolved existing stress gains suggests that GC functions as a stress modifier, which is in contrast to the classical conception that GC is a stress mediator. Although when and how GC functions as a stress modifier need to be studied in more detail, we speculate that repeated short-term stress can restore stress coping ability, presumably by repeatedly boosting the feedback and feedforward regulatory mechanisms of the HPA axis, which was disrupted in CRST mice (Fig. 10). This possibility does not conflict with the classical conception that chronically imposed GC produces cumulative effects on stress gains due to the points described below. It will be worth studying whether the circadian oscillation of basal GC levels[2,45], which is disrupted in patients with depression[40,41], functions as a stress modifier by daily resetting the stress coping system. In healthy individuals, the basal serum GC levels vary throughout the day, with the highest in the early

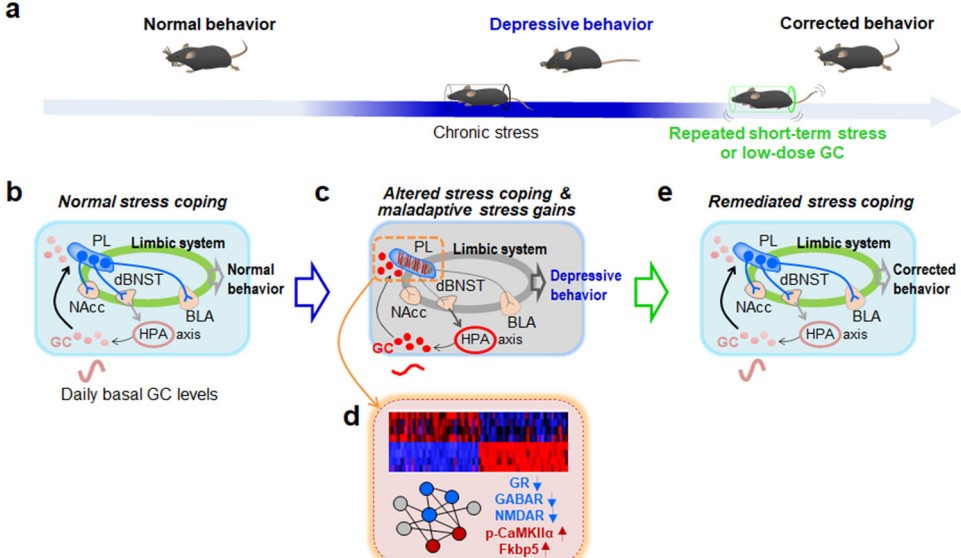

**Fig. 10 A summary and hypothetical model for the recovery of maladaptively accumulated stress gains by treatment with short-term behavioral stress or low-dose GC. a** Chronic stress produces maladaptive changes in the brain and persistent depressive behavior. Repeated treatment with short-term behavioral stress or low-dose GC reverses the stress-induced maladaptive changes and rescues depressive behavior. **b** Diagram showing the brain with the normal activity of PL outputs (PL→NAcc, PL→dBNST, and PL→BLA) and the limbic system (green), and normal activity of the HPA axis including the basal GC release at a normal circadian cycle and its feedforward effects on the PL. **c, d** Chronic stress overstimulates the PL primarily due to increased GC (**c**), which results in genome-wide gene expression alteration (**d**). As a result, the neural activity of PL outputs (PL→NAcc, PL→dBNST, and PL→BLA) and those of the associated limbic system are altered, and the basal GC release is enhanced and trailed off from a normal circadian cycle (**c**). Heatmap presents stress-induced genomic changes and their reversal by RS5. The expression levels of GR, GABAR, NMDAR, p-CaMKIIα, and Fkbp5 in the PL are up- or down-regulated after chronic stress (**d**), and their physiological effects on depressive behavior are characterized in the present study. **e** Repeated treatment with short-term behavioral stress or low-dose GC reverses the disrupted neural activity of PL neurons, the altered gene expression profiles, and the altered activity of the HPA axis including GC release, and rectifies the impaired feedforward effects of GC on the PL.

morning and then falling throughout the day to the late evening[2,45]. Second, repeated treatment with short-term behavioral stress or repeated treatment with GC could be used as an anti-depressive strategy. Considering that RS5, but not RS10 and RS15, produced anti-depressive effects, the behavioral method has a relatively narrow window to afford therapeutic effects. However, if provided by properly designed and implemented based on further studies, the behavioral method should have a privilege over the pharmacological methods in certain conditions. On the other hand, challenging with GC might provide versatility to resolve existing stress gains. The profound therapeutic effects of short-term behavioral stress and exogenous GC demonstrated in this study warrant further investigation for the detailed mechanisms and their utilities. Third, the finding that repeated short-term stress or repeated treatment with GC did not strengthen existing stress gains but could resolve them (Figs. 1 and 2) raises the possibility that prior stress gains might become transiently deconsolidated and exist in a temporarily labile state upon every new stress inputs generated during chronic stress phase. It will be worth studying the detailed mechanisms as to how and when accumulated prior stress grains become strengthened or deconsolidated and resolved after challenging with new stress inputs.

Concerning GC-dependent changes, p-CaMKII and its associated factors in the PL appear to mediate CRST and RS5 effects. Chronic stress increased p-CaMKII levels in PL neurons (Fig. 6a–i), whereas molecular or pharmacological inhibition of CaMKIIα in the PL produced anti-depressive effects (Figs. 6l–t and 7b–g), and p-CaMKII levels were negatively correlated with GR expression in PL neurons (Fig. 6f–q). siRNA-mediated inhibition of GR in the PL increased Fkbp5 expression and produced depressive-like behavior (Fig. 5o–u). Fkbp5, in coordination with Hsp90, negatively regulates GR nuclear translocation[46]. In these molecular networks regulating PL neurons, RS5 treatment restored not only the stress-induced increased p-CaMKII levels in PL neurons but also the reduced expression of GR and increased expression of Fkbp5 in the PL (Figs. 5 and 6). These results suggest that p-CaMKII, GR, Fkbp5, and their associated molecular networks (Fig. 5) are critical players regulating PL neurons and that RS5 produces the remediating effects by targeting the neural and molecular mechanisms activated by chronic stress.

Chronic stress adaptively modifies the neural activity of the mPFC. Glutamate/glutamine levels and the neural activity of glutamatergic neurons in the mPFC are reduced in CUMS-, CRST-, and CSDS-induced depression models in mice[47–49]. Indeed, stimulation of PL neurons in CRST mice suppressed depressive-like phenotypes (Supplementary Fig. 8). Conversely, chemogenetic inhibition of PL neurons during RS5 treatment dissipated the anti-depressive effects of RS5 (Fig. 8). Furthermore, chemogenetic inhibition of the PL→ dBNST, PL→ BLA, or PL→ NAcc circuits during RS5 treatment abolished the anti-depressive effects of RS5 (Fig. 9). Previous studies also reported that activation of the PL[50], the PL→ NAcc circuit[51], and the PL→ BLA circuit[52] promotes anti-depressive effects. In CRST mice, the neural activity of PL neurons projecting to dBNST, NAcc, and BLA was decreased, whereas RS5 treatment reversed the decreased activity of the PL neural networks. Interestingly, however, the PL→ dBNST circuit mediated the remediating effects of RS5 on the basal GC levels, whereas the PL→ BLA and PL→ NAcc circuits did not (Fig. 9). It will be worth investigating the distinct role of the PL→dBNST circuit, relative to those of the PL→ BLA and PL→ NAcc circuits, in regulating stress coping adaptability.

Our c-Fos mapping analyses indicated that the IL, vSub, and many other regions had c-Fos expression following RS5 or low-dose CORT treatment (Fig. 4a–c and Supplementary Data 1). The

IL and PL constitute the core of the mPFC that regulates stress coping responses and affective behavior[11,53]. The vSub is also involved in stress, emotion, and affective behavior[54,55]. The results of the present study indicate that PL neurons and their associated neural systems, including those projecting to dBNST, BLA, and NAcc, compose the critical neural nodes and edges that support stress-induced depressive behavior and RS5 effects, which is summarized in Fig. 10b–e. It is possible that the IL, vSub, and other regions of the limbic system could also play a role in stress coping adaptability regulated by RS5 and low-dose CORT.

Stress inoculation is a pretreatment strategy that improves subsequent stress coping and emotional regulation[56]. Stress inoculation in mice, by placing them behind a mesh-screen barrier in a cage containing an aggressor mouse for 15 min, enhances subsequent stress-coping behavior and cognitive function[57]. Another form of stress inoculation, called predictable chronic mild stress (PCMS), uses 5-min restraint for 28 days to improve mood, hippocampal neurogenesis, and memory in rats[58]. In those studies, stress inoculation or PCMS is regarded as an immunization trick to enhance the coping ability for future stress[56]. PCMS treated for 28 days induced anti-depressive effects in rats[58]. In contrast, our RS5 treatment (daily 5-min restraint for 14 days) in normal mice did not induce depressive behavior (Fig. 1a–g). Therefore, it will be worth studying that a pretreatment paradigm of daily 5-min restraint for 7–14 days in normal mice would also produce resiliency to future chronic stress. It is possible that stress inoculation/PCMS and RS5 could have a certain common neural mechanism. Nonetheless, our experimental procedure provoking short-term stress deals with a therapeutic strategy, whereas stress inoculation/PCMS is preventative.

In the present study, we used the sociability test, sucrose preference test, TST, and FST to read out behavioral changes of mice induced by CRST and RS5 treatment. Recently, it has been proposed that the FST behavior does not model human depression, but it measures stress coping behavior[59]. Regarding this view, the behavioral changes measured by the FST could be interpreted to reflect stress coping adaptability.

## Methods

**Animals.** Seven-week-old male C57/BL6 mice were purchased from Daehan Bio-Link (Eumseong, Chungbuk, Republic of Korea). Eight-week-old male and female ICR (CD1) mice were purchased from OrientBio. Inc. (Seongnam, Gyeonggi, Republic of Korea) and used as breeders. All mice were housed in pairs in standard clear plastic cages in a temperature (23–24 °C)- and humidity (50–60%)-controlled room, with food and drinking water available ad libitum. The animal room was maintained on a 12-h light/dark cycle (light on at 7 a.m.) in a specific pathogen-free environment. Animal experiments were performed in accordance with the animal care guidelines of the Animal Ethics Committee of Ewha Womans University, and all protocols used in the present study were approved by the Institutional Animal Care and Use Committee (IACUC) of Ewha Womans University (IACUC16-018 and IACUC19-015).

**Short-term restraint stress.** Mice were individually placed in a 50-ml polypropylene conical tube with many ventilation holes and restrained for 5, 10, 15 min, or the indicated time beginning at 10 a.m., and this treatment was repeated for the indicated number of days. After each restraint session, the mice were returned to their home cages.

**Chronic restraint stress (CRST).** CRST was carried out as described previously in ref. [60]. Mice were individually restrained in a well-ventilated 50-ml conical tube for 2 h daily from 10 a.m. to 12 p.m., and this procedure was repeated for 14 days. After each daily restraint session, the mice were placed in their home cages with free access to food and water.

**Chronic social defeat stress (CSDS).** CSDS treatment and selection of stress-susceptible mice were carried out as described previously in ref. [61]. C57/BL6 mice were individually exposed to a novel ICR aggressor for 10 min to produce physical defeat stress and then housed for the remainder of the day in a compartment of the ICR aggressor's cage partitioned with a perforated acrylic divider. This procedure was repeated for 10 days with a different ICR aggressor each day. ICR aggressors

were preselected based on the following criteria: the latency to initial attack was less than 60 s but higher than 30 s, and any aggressors did not exhibit overly aggressive attacks such as severe wounding and bleeding during social defeat exposure, and if they did not meet the selection criteria, they were excluded from the next experiment.

Stress-susceptible and resilient mice were selected on the basis of the social interaction ratio of individual animals assessed by the social interaction test on day 11. CSDS-treated mice were individually placed in an open field (45 cm ×45 cm × 40 cm) containing an empty perforated interaction box placed at one side of the open field and allowed to explore the open field for 2.5 min. The time spent exploring the empty interaction box in the field was recorded. While the subject mouse was returned to its home cage for 1 min, an ICR aggressor was placed in the interaction box, and then the subject mouse was allowed to freely explore the open field for 2.5 min. The time spent interacting with the ICR aggressor in the interaction box in the field was recorded. The social interaction (SI) ratio was defined as the value of the total time spent in the interaction zone with the ICR aggressor divided by the time spent in the interaction zone when the ICR aggressor is absent. When the SI ratio was below 1.0, mice were grouped as susceptible, whereas the SI ratio was 1.0 and above, mice were grouped as resilient. Susceptible CSDS mice were used for further study.

**Maternal stress (MS) and postnatal stress.** Male and female ICR mice at 8–9 weeks of age were crossed, and pregnant females were randomly assigned to the maternal stress (MS) group or normal (N) control group. MS pregnant females were treated with 2-h restraint daily for 2 h (10 a.m.–12 p.m.) from 8.5 days post coitus (dpc) to delivery (at 19.5–20.5 dpc). Normal (N) pregnant females were maintained without the stress procedure in parallel to the MS females. The off-spring of both groups were weaned at postnatal day 20 (PN20) and reared in pairs with the same sex in cages under standard conditions. For mouse groups assigned for CRST treatment, on PN49, offspring from N control mothers or MS mothers were separately housed in same-sex pairs from different litters to prevent possible litter effects. Both N and MS mice were then exposed to daily 2-h restraint for 14 days (during PN50–PN63). Afterward, half of the CRST-treated mice were subjected to the RS5 regimen, and the remaining half were used as the CRST control group.

**Immunohistochemistry.** Immunohistochemistry was performed as described previously[38,39]. Mice were perfused with 4% paraformaldehyde via the transcardiac method and isolated brains were postfixed at 4 °C overnight. Each brain was coronally sectioned into 40-μm thicknesses using a vibratome (VT1000S, Leica Instruments, Nussloch, Germany). The collected sections were blocked for 1 h with 4% bovine serum albumin in phosphate-buffered saline (PBS) containing 0.1% Triton X-100 and then incubated with primary antibody at 4 °C overnight. After washing, sections were reacted for 90 min with biotinylated secondary antibody: anti-rabbit IgG (BA-1000, Vector Laboratories, Burlingame, CA, USA) or anti-mouse IgG (BA-9200, Vector Laboratories) diluted at 1:200 in PBST. Signals were visualized using an ABC Elite kit (PK-6200, Vector Laboratories).

For the analysis of c-Fos expression induced by short-term restraint stress, CRST mice were treated, on post-stress day 8, with a single 5-min restraint or 15-min restraint (S5 × 1d and S15 × 1d, respectively) and sacrificed, 20 and 10 min later, respectively. Other groups of CRST mice were treated with daily 5-min restraint or 15-min restraint for 7 days and sacrificed 20 and 10 min after the last 5-min restraint or 15-min restraint, respectively, on post-stress day 8 (S5 × 8d and S15 × 8d, respectively).

For the analysis of c-Fos expression induced by low-dose CORT injection, CRST mice were given CORT injections at 0.1, 0.5, or 1.0 mg kg$^{-1}$ per day for 7 days and sacrificed 25 min after an additional CORT injection on post-stress day 8.

The level of c-Fos expression in specific brain regions was quantified using an Olympus BX 51 microscope equipped with a DP71 camera and MetaMorph Microscopy Automation and Image Analysis software (Molecular Devices, Sunnyvale, CA, USA) and also using a six-point rating scale, as described previously[62]. The c-Fos expression level was assessed by a numerical grade of 0–5 scales; 0 assigned for 0–50 c-Fos-positive cells/mm$^2$ in a counting region; +1 for 51–100 c-Fos-positive cells/mm$^2$; +2 for 101–200 c-Fos-positive cells/mm$^2$; +3 for 201–350 c-Fos-positive cells/mm$^2$; +4 for 351–600 c-Fos-positive cells/mm$^2$; and +5 for >600 c-Fos-positive cells/mm$^2$. The average c-Fos expression score in each region was rounded to the nearest tenth.

The primary antibodies used were anti-c-Fos (sc-52, 1:300 for immunofluorescence, 1:2000 for immunohistochemistry; sc-271243, 1:300, Santa Cruz Biotechnology, CA, USA), anti-Glu4 (G9282, Sigma-Aldrich, MO, USA), anti-GAD67 (MAB5406, EMD Millipore, CA, USA), anti-GR (sc-136209, 1:500, Santa Cruz Biotechnology; 12041 S, 1:500, Cell Signaling Technology, MA, USA), anti-Fkpb5 (GTX84491, 1:50, Genetex, CA, USA), anti-NR1 (sc-1467, 1:50, Santa Cruz Biotechnology), anti-NR2A (sc-9056, 1:50, Santa Cruz Biotechnology), anti-NR2B (ab93610, 1:100, Abcam, Cambridge, UK), anti-p-CaMKIIα (Thr286) sc-12886, 1:1000, Santa Cruz Biotechnology), and anti-p-ERK1/2 (Thr202/Tyr204) (4370 s, 1:500, Cell Signaling Technology). For immunofluorescence staining, the following fluorescence-tagged secondary antibodies were used: anti-rabbit IgG DyLight488 (DI-1488; 1:500, Vector Laboratories), anti-rabbit IgG DyLight594 (DI-1094; 1:500, Vector Laboratories), anti-mouse IgG DyLight488 (DI-2488; 1:500, Vector Laboratories), anti-mouse IgG DyLight594 (DI-

2594; 1:500, Vector Laboratories), and anti-goat IgG-FITC (sc-2356, 1:200, Santa Cruz Biotechnology). Stained sections were mounted with an antifade mounting solution containing DAPI (H-1200; Vector Laboratories). Stained immunofluorescence images were analyzed using an Olympus BX 51 microscope equipped with a DP71 camera and the MetaMorph program (Molecular Devices).

**Real-time PCR analysis.** Real-time PCR was carried out as described previously[38,39]. Total RNA was purified from brain tissue using TRIzol reagent (15596018, Invitrogen Life Technologies, Carlsbad, CA, USA) and was treated with DNase I (M610A, Promega, MO, USA) to eliminate genomic DNA contamination. Reverse transcription of total RNA (2 μg) was conducted in a volume of 20 μl using a reverse transcription system (A3500, Promega, MO, USA). Real-time PCR was performed with 10 μl of 2X iQ™ SYBR Green Supermix (#170882, Bio-Rad Laboratories, Foster City, CA, USA), 1 μl each of 5 pmol/μl forward and reverse primers, and 4 μl of complementary DNA (cDNA) (1/8 dilution of the conversion) in a total volume of 20 μl using a CFX 96 Real-Time PCR System Detector (Bio-Rad Laboratories). The Bio-Rad CFX Manager 3.1 was used to analyze qPCR data. The primer sets used in PCR were listed in Supplementary Data 5.

**Corticosterone measurement.** Serum corticosterone levels were measured as described previously[39]. Mice were anesthetized by intraperitoneal injection (300 μL per mouse) with 2.5% avertin (2,2,2-tribromoethanol; 300 mg kg$^{-1}$) (T48402, Sigma-Aldrich) in tert-amyl alcohol (24,048-6, Sigma-Aldrich) diluted to 2.5% avertin with saline. Blood was collected from the abdominal aortas of sacrificed mice in the morning (8 a.m.–12 p.m.), and serum was obtained by centrifugation at 3000×g and 4 °C for 15 min and then stored at −80 °C until use. Each diluted serum sample (10 μl) was mixed on a 96-well plate with an equal volume of a steroid displacement reagent solution and sera diluted at 1: 40 or 1:80 in the ELISA assay buffer provided in the corticosterone ELISA kit (ADI-901-097, Enzo Life Sciences, NY, USA). The reaction was incubated for 2 h at room temperature on an orbital shaker rotating at 120 rpm. The reaction mixture in each well was discarded, and the plate was rinsed with washing buffer. The pNpp substrate solution provided in the kit was added at 200 μl/well, and the plate was incubated for 1 h at room temperature without shaking. Finally, the absorbance at 405 nm was measured using a spectrofluorometer (SpectraMax® M5, Molecular Devices, Sunnyvale, CA, USA) and analyzed using SoftMax Pro software version 5.4. Corticosterone levels were analyzed using an online data analysis tool for ELISA of Enzo Life Science (http://www.myassays.com).

**Chemogenetic modulation of the activity of specific neurons.** Chemogenetic manipulation of neurons was carried out using a DREADDs system as described previously[63]. Mice were anesthetized with a mixture of ketamine and xylazine (3.5:1). To activate PL neurons, the CaMKIIα promoter-driven hM3D viral vector AAV8-CaMKIIα-hM3D(Gq)-mCherry (3 × 10$^{10}$ viral particles ml$^{-1}$) or the hSyn promoter-driven hM3D viral vector AAV8-hSyn-hM3D(Gq)-mCherry (4.3 × 10$^{10}$ viral particles ml$^{-1}$) was bilaterally injected into the PL using a stereotaxic injection system (Vernier Stereotaxic Instrument, Leica Biosystems, Wetzlar, Germany) and a Hamilton syringe with a 30-gauge needle. To inhibit PL neurons, the CaMKIIα promoter-driven hM4D viral vector AAV8-CaMKIIα-hM4D(Gi)-mCherry (1.6 × 10$^{10}$ viral particles ml$^{-1}$) was bilaterally injected into the PL. After 7 recovery days, mice were exposed to CRST. CRST mice were injected with vehicle or clozapine N-oxide (CNO) (0.1, 1, or 3 mg kg$^{-1}$) 30 min before treatment with 5-min restraint according to the experimental design.

To transduce hM4D(Gi) in the PL→dBNST circuit, the viral vector AAV8-hSyn-DIO-hM4D(Gi)-mCherry (3 × 10$^{10}$ viral particles ml$^{-1}$) was bilaterally injected into the PL (AP, +2.0; ML, ±0.23; DV, −2.0 mm), and the retrograde Cre vector AAV8-EF1α-mCherry-IRES-WGA-Cre (1.5 × 10$^{10}$ viral particles ml$^{-1}$) was injected into the dBNST (AP, +0.25; ML, ±1.15; DV, −4.2 mm) as depicted in the experimental design. For chemogenetic inhibition of PL→dBNST neurons, they were injected with vehicle or CNO (3 mg kg$^{-1}$, i.p.) 30 min before treatment with 5-min restraint, as depicted in the experimental design. The injection volume and injection rate for each target site were as follows: for PL, 0.5 μl at a rate of 0.25 μl per min; and for dBNST, 0.3 μl at a rate of 0.15 μl per min. The injection needle was left in the site for 5 min to ensure adequate delivery and was withdrawn slowly over the course of 5 min.

To transduce hM4D(Gi) in the PL→NAcc or PL→BLA circuit, the viral vector AAV8-hSyn-DIO-hM4D(Gi)-mCherry (3 × 10$^{10}$ viral particles ml$^{-1}$) was bilaterally injected into the PL (AP, +2.0; ML, ±0.23; DV, −2.0 mm), and the retrograde Cre vector AAV8-EF1α-mCherry-IRES-WGA-Cre (1.5 × 10$^{10}$ viral particles ml$^{-1}$) was injected into the NAc (AP, +1.65; ML, ±1.15; DV, −5.15 mm) or BLA (AP, −1.3; ML, ±3.35; DV, −5.2 mm) as depicted in the experimental design.

AAV8-CaMKIIα-hM3D(Gq)-mCherry, AAV8-CaMKIIα-hM4D(Gi)-mCherry, AAV8-hSyn-hM3D(Gq)-mCherry, and AAV8-hSyn-DIO-hM4D(Gi)-mCherry were obtained from Dr. Bryan Roth (Addgene: plasmids #50476, #50477, #50474, and #44362, respectively). AAV8-EF1α-mCherry-IRES-WGA-Cre was purchased from UNC Vector Core (University of North Carolina, NC, USA). Viral vectors were aliquoted upon arrival and stored at −80 °C until use. Each viral vector was

diluted in 1x PBS (13.7 mM NaCl, 0.27 mM KCl, 0.8 mM $Na_2HPO4$, 0.2 mM $KH_2PO_4$, pH 7.4).

**Neural circuit mapping with antero- and retrograde tracers.** Mice were anesthetized with ketamine and xylazine (3.5:1). To label the anterograde projections of PL neurons, AAV2/1-CaMKIIα-EYFP ($1.0 \times 10^9$ viral particles $ml^{-1}$) (AV-1-PV1917, University of Pennsylvania, PA, USA) was stereotaxically injected into the PL (AP, +2.0; ML, −0.23; DV, −2.0 mm) with a volume of 0.5 μl at the rate of 0.25 μl/min using a stereotaxic injection system (Vernier Stereotaxic Instrument) and a 30-gauge needle. Two weeks after the stereotaxic injection, the brains were prepared after perfusion with 4% paraformaldehyde via a trans-cardiac method. To label the neuronal afferents to the dBNST, Alexa Fluor 488-conjugated cholera toxin subunit B (CTB) (0.1% weight per volume) (C34755, Molecular Probes, Eugene, OR, USA), a retrograde tracer, was stereotaxically injected into the dBNST (AP, +0.25; ML, −1.15; DV, −4.2 mm) in a volume of 0.3 μl at a rate of 0.15 μl per min using a 30-gauge needle. One week after the stereotaxic injection, the brains were prepared after perfusion with 4% paraformaldehyde as described above. The brains were coronally sectioned into 40-μm thicknesses with a vibratome (Leica VT 1000 S; Leica Instruments, Nussloch, Germany). Fluorescence images were collected and analyzed using an Olympus BX 51 microscope equipped with an X-cite 120 fluorescence illuminator (EXFO Life Science & Industrial Division, Ontario, Canada), a DP71 camera (Olympus), and MetaMorph Microscopy Automation and Image Analysis software (Molecular Devices, Sunnyvale, CA, USA).

**Optogenetic manipulation of the PL-to-NAcc pathway.** Optogenetic manipulation was performed as described previously[64]. The viral vector AAV2/1-CaM-KIIα-eNpHR-EYFP ($1.15 \times 10^9$ viral particles $ml^{-1}$) or AAV2/1-CaMKIIα-EYFP ($1 \times 10^9$ viral particles $ml^{-1}$) was stereotaxically injected into the PL (AP, +2.0; ML, +0.23; DV, −2.0 mm) at the right side in a volume of 0.5 μl at a rate of 0.25 μl per min using a stereotaxic injection system (Vernier Stereotaxic Instrument) and a 30-gauge needle. After 7 recovery days, an optic fiber (200 μm in diameter, 5.15 mm in length; B280-2508-5, Doric Lenses, Quebec, Canada) was implanted into the NAc (AP, +1.65; ML, +1.15; DV, −5.15 mm) at the right side in normal mice. For optical inhibition of PL-to-NAcc neurons, the implanted optic fiber was connected to the 532-nm diode-pumped solid-state (DPSS) yellow laser (GL532T3-100; Shanghai Laser & Optics Century, Shanghai, China) through a fiber-optic cannula with FC/PC adapter (D202-2027-5; Doric Lenses). In the behavioral tests, the ferrule of a fiber-optic cannula was connected to a patch cord using a zirconia split sleeve (F210-3002, Doric lenses) and a fiber-optic rotary joint (B300-0010, Doric lenses). The yellow light intensity was set at 4 mW. Yellow light pulses were generated using a stimulator (#33521 A, Agilent Technologies, Santa Clara, CA, USA) and were delivered at 20 Hz with a spike width of 30-ms with a time interval (3-min for sociability test and 2-min for TST or FST). The light intensity through the optic fiber was confirmed with a light detector (PM200; Thorlabs, Newton, NJ, USA) equipped with a light sensor (S121C, Thorlabs) before starting the behavioral test.

The AAV viral vectors AAV2/1-CaMKIIα-eNpHR-EYFP (AV-1-26971P) and AAV2/1-CaMKIIα-EYFP (AV-1-PV1917) were purchased from Penn Vector Core (University of Pennsylvania, PA, USA).

**Gene knockdown using small interfering RNA (siRNA).** Stereotaxic injection of siRNA was carried out as described previously[38,64]. Mice were anesthetized with a mixture of ketamine hydrochloride (50 mg $ml^{-1}$) and xylazine hydrochloride (23.3 mg $ml^{-1}$) (3.5: 1) at a dose of 2.5 μl $g^{-1}$ body weight. One volume of diluted (50 ng $μl^{-1}$) siRNA was mixed with 2.5 volumes of Neurofect transfection reagent (T800075, Genlatis, San Diego, CA, USA) and 0.5 volumes of 50% sucrose and incubated for 20 min before injection into the brain. A total of 0.5 μl of a mixture containing 5.94 ng of siRNA-GR, siRNA-CaMKIIα, siRNA-ERK1, siRNA-ERK2, or siRNA-control was stereotaxically injected into each side of the PL (AP, +2.0; ML, ±0.23; DV, −2.0 mm) at a rate of 0.25 μl per min using a stereotaxic injection system (Vernier Stereotaxic Instrument) and a 30-gauge needle. Brain tissues were prepared for the analyses of knockdown or gene expression changes 48–72 h after siRNA injection. Behavioral tests were performed between 48 and 72 h after the injection of siRNA.

The siRNAs used were siRNA-control (SN-1012), siRNA-CaMKIIα (#12322, NM_177407.2, NM_009792.1), siRNA-MAPK3 (ERK1) (#26417, NM_011952.2), siRNA-MAPK1(ERK2) (#26413, NM_011949.3, NM_001038663.1), or siNr3c1(GR) (#14815, NM_008173.3), all of which were purchased from Bioneer Co. (Daejeon, Korea).

**Drug infusion through a cannula.** Cannula implantation and drug infusion were performed as described previously[64]. Mice were anesthetized with ketamine and xylazine (3.5: 1). A 26-gauge guide cannula (C315G/SPC, Plastics One, Bilaney, UK) was stereotaxically implanted into the PL (AP, +2.0; ML, +0.23; DV, −2.0 mm). The cannula was fixed in place by applying Super-Bond C&B™ dental adhesive (Sun Medical, Moriyama, Japan) to the surface of the mouse skull around the cannula base. After 15 min, dental cement (Vertex-Dental, AA Zeist, The Netherlands) was applied to fix the cannula to the skull surface and then sealed with a dummy cannula (C313DC, Plastics One). The mice had more than 7

recovery days. While mice were anesthetized with 1.4% isoflurane in a gas mixture of 70% nitrous oxide and 30% oxygen, drugs were infused into the PL in a volume of 0.5 μl via a 33-gauge internal cannula (C315I, Plastics One). KN62 (2.5 nmol/ injection; I2142, Sigma-Aldrich) was delivered into the PL in a volume of 0.5 μl at a rate of 0.25 μl per min through cannulation.

**Drug administration.** Imipramine (I0899, Sigma-Aldrich), corticosterone (C2505, Sigma-Aldrich), NBI27914 hydrochloride (1591, Tocris Bioscience), RU486 (M8046, Sigma-Aldrich), and Clozapine N-oxide (CNO) dihydrochloride (6329, Tocris Bioscience) were diluted in 0.9 % saline and injected intraperitoneally in a volume of 120 μl.

**Western blot analysis.** Western blot analyses were performed as described previously[38]. PL tissue from three to four animals was homogenized in RIPA buffer (50 mM Tris-HCl, pH 7.4, 150 mM NaCl, 0.5% Nonidet P-40, 0.25% sodium deoxycholate, 1 mM EDTA, and 0.5% Triton X-100) containing a protease inhibitor cocktail (11836153001, Roche, Mannheim, Germany). Tissue homogenates loaded at 20 μg of protein per lane were resolved on SDS-PAGE and transferred onto PVDF (1620177, Bio-Rad Laboratories). The membranes were incubated with blocking solution (5% skim milk; 232100, Difco, MI, USA) in Tris-buffer based saline containing Tween 20 (TBST; 150 mM NaCl, 50 mM Tris-HCl, pH 7.4, 0.1% Tween 20) followed by primary antibody diluted in 1% skim milk at 4 °C overnight. After washing with 1x TBST six times for 10 min each time, the membranes were incubated with a secondary antibody for 1 h. The secondary antibodies used were anti-rabbit IgG-HRP (sc-2004, Santa Cruz Biotechnology) and anti-mouse IgG-HRP (sc-2005, Santa Cruz Biotechnology) diluted 1:2000 in 1% skim milk. After washing, specific signals on the blots were visualized using an enhanced chemi-luminescence kit (ELPIS-Biotech, Daejeon, Korea). Western blot images were quantified using Image J analysis software v1.51k (NIH Image, Bethesda, MD, USA).

The primary antibodies were anti-p-CaMKIIα (Thr286) (sc-12886, 1:2000, Santa Cruz Biotechnology), CaMKIIα (A-1) (sc-13141, 1:2000, Santa Cruz Biotechnology), anti-p-ERK1/2 (Thr202/Tyr204) (4370 s, 1:500, Cell Signaling Technology), anti-ERK1/2 (MK1) (sc-135900, 1:2000, Santa Cruz Biotechnology), and anti-β-actin (C4) (sc-47778, 1:3000, Santa Cruz Biotechnology).

**Adrenalectomy.** Adrenalectomy (ADX) was carried out as described previously[65]. Mice were anesthetized using a mixture of ketamine and xylene (3.5:1). Surgical tools were presterilized with Bromosept (DAESUNG Microbiological Labs. Co., Kyonggi, Republic of Korea). Mice were placed prone on the surgical bed. The dorsal side of the body was covered with a 10% of povidone-iodine solution (#261, Sungkwang, Chungnam, Republic of Korea). The left side of the back was grabbed with forceps (#5, Fine Science Tools Inc., North Vancouver, B.C. Canada), and the skin and peritoneum were sequentially incised with micro-dissecting scissors (S3271, Sigma-Aldrich). Then, the adrenal gland was gripped with forceps (#5, Fine Science Tools Inc.) and cut off with a scalpel blade (S2646, Sigma-Aldrich) assembled on a scalpel handle (S2896, Sigma-Aldrich). The peritoneum and skin were sequentially stitched with silk suture 4/0-18 mm (SK434, Ailee Company Limited, Busan, Republic of Korea). The right side operation followed the same procedure as the left. Adrenalectomized mice were subsequently given 0.9% saline solution instead of drinking water, but they were given no corticosterone replacement. Sham-operated mice underwent the same surgical procedure: the adrenal glands were grabbed with the forceps, but they were not removed. After 7 recovery days, half of the ADX and sham-operated mice were subjected to RS5 treatment, and the remaining mice served as controls.

**Microarray analysis.** Microarray analysis was carried out as described previously[38]. Two independent sets of control (CON), CRST, and CRST + RS5 groups were prepared. Mice were subjected to CRST treatment followed by RS5 and then sacrificed on post-stress day 22. Total RNA was extracted from the pooled PL tissue of 6–7 animals in each group using Trizol (Invitrogen Life Technologies), followed by purification using RNeasy columns (Qiagen, Valencia, USA). The purified RNA had an A260/280 ratio of 1.9–2.1 as determined by an ND-1000 Spectrophotometer (NanoDrop, Wilmington, DE, USA) and an Agilent 2100 Bioanalyzer (Agilent Technologies, Palo Alto, CA, USA).

Total RNA (550 ng for each) was converted to first- and second-strand cDNA by in vitro transcription. The cDNA was then used to synthesize biotin-labeled cDNA samples. Purified biotin-labeled cDNA samples were fragmented by heating them to 94 °C for 35 min in an array fragmentation buffer. Each of fragmented and biotin-labeled cDNA samples (750 ng) was hybridized to the mouse Ref8 expression v.2 bead array (Illumina, Inc., San Diego, USA) for 16–18 h at 58 °C, according to the manufacturer's instructions. After washing, the array signals were amplified with Amersham Fluorolink streptavidin-Cy3 (GE Healthcare Bio-Sciences, Little Chalfont, UK) and scanned with an Illumina Bead Array Reader, a confocal scanner according to the manufacturer's instructions.

The quality of hybridization and overall chip array performance, which included the internal quality control probes and the scanned raw data extracted using the Illumina GenomeStudio v2011.1 software (Gene Expression Module v1.9.0) were monitored by Macrogen Inc. (Seoul, South Korea). The microarray

signals were converted into log2 scale values and normalized by the quantile method.

Gene expression profiles were analyzed using Multiexperiment Viewer (MeV), version 4.9 (http://mev.tm4.org/). Genes whose expression were up- or down-regulated ≥1.2-fold by CRST (CON vs. CRST) and those changed ≥1.2-fold after RS5 treatment of CRST-treated mice (CRST vs. CRST + RS5) were selected. The expression profiles of the 264 CRST-responsive genes were presented from the highest to the lowest value in a heatmap followed by unsupervised alignment with the respective genes from the CRST vs. CRST + RS5 comparison. Independently, the gene expression profiles of the 722 RS5-responsive genes were aligned similarly as above in a heatmap and then overlaid with the respective genes from the CON vs. CRST comparison.

**Gene Ontology (GO) enrichment analysis**. GO enrichment analysis was carried out using the STRING (Search Tool for Retrieval of Interacting Genes/Proteins) v11 database[32] (available at: http://string-db.org/). Of the total 896 (264 + 722) genes selected as described in the previous section, 860 genes were assigned to functional groups with biological process (BP) terms, whereas the 36 genes with an unspecified identity were excluded from further analyses. The GO term hierarchy was assigned based on the Mouse Genome Database[66] (http://www.informatics.jax.org).

Serial $K$-means clustering was used to group the 860 genes into multiple functional clusters. Each increase in $K$ value added a new cluster, which contained members mostly supplied from the largest cluster (cluster 1). When >75% of the genes in a cluster remained as a cluster in the next serial clustering, those two inter-clusters were considered to be in the same functional group and are therefore marked with the same color code. The retention rate of most cognate inter-clusters was higher than 85% for grouping into 5–10 clusters ($k = 5$–10), whereas the retention rate dropped after grouping into 11 or more clusters.

A functional protein–protein interaction (PPI) network (PPI enrichment $p$ value, <1.0e-16) consisting of eight clusters ($k = 8$) was chosen as a representative grouping: Cluster 1 contained the most members (622 genes). However, the members of cluster 1 were poorly interactive and scattered outside the functional networks formed by the remaining seven clusters, which included 238 genes in total. Therefore, cluster 1 was excluded from further analyses. Clusters 2 to 8 contained 71, 39, 29, 28, 27, 25, and 19 genes, respectively. Clusters 5 and 7, which contained 28 and 25 genes, respectively, covered the GO terms "response to stress" and "response to glucocorticoids", along with the cell communication, the establishment of localization, and response to endogenous stimuli.

Potential interactors which were missed from the microarray screening but that could exist in the functional PPI networks were sought in the STRING network database. Using edge confidence of 0.7 and higher to classify nodes in the PPI networks, that effort produced 16 new interactors. Afterward, functional networks were constructed to include the members of clusters 5 and 7 and the 16 new members. The nodes and edges for interactions with members were color-coded using Cytoscape StringApp v3.7.2[67].

Real-time PCR was used to examine the expression profiles of the Nr3c1(GR), Nr3c2(MR), Fkbp5, Fkbp4, Hsp90aa1, and Hsp90ab1 genes, which were arbitrarily chosen from cluster 7, and the expression of Dusp1, CaMKIIα, Mapk3(ERK1), and Mapk1(ERK2), which were selected from cluster 5 for the indicated groups.

**Behavioral assessments**. Behavioral tests were carried out as described previously[38,39]. All behavioral tests were recorded with a computerized video tracking system (SMART; Panlab S.I., Barcelona, Spain) or a webcam recording system (HD Webcam #C210; Logitech, USA). All behavioral tests among groups and within groups were conducted in a randomized fashion or in an alternating manner with respect to test order and position within the testing equipment or the test field (e.g., left vs. right side, between positions in the test field). CRST-induced persistent behavioral changes and RS5 effects were verified in a blind manner by two experimenters. All animals were housed in pairs unless otherwise indicated. No bedding change was made the day before the behavioral tests. The behavioral testing room was lit with 20 lux of indirect illumination for the sociability test and 250 lux for the TST and FST. Mice were given 20–30 min to adapt to the behavior testing room prior to starting each behavioral test. The background sound in the testing room was masked with 65 dB of white noise throughout all behavioral tests. All behavioral tests were performed from 9 a.m.–3 p.m. during the light cycle. After each behavioral test, all parts of the apparatus were cleaned with 70% ethanol.

**Two-chamber social interaction test**. The two-chamber social interaction test (SIT) was developed as a modified version of the three-chamber test[68] and the U-shaped two-choice field test[38,39]. The two-chamber apparatus consisted of two symmetrical chambers (26 cm × 26 cm rectangular floor with 40 cm walls) separated by an intermediate path (8 cm wide × 10 cm long × 26 cm high), with an empty circular grid cage (12 cm in diameter × 33 cm in height) in each chamber. A subject mouse was placed in the middle path and allowed to freely explore the two chambers for 5 min as habituation. While the subject mouse was returned to its home cage for 2 min, a social target (C57BL/6) was placed in one of the grid cages. As soon as the social target was stabilized, the subject mouse was placed in the middle of the two choice chambers, with a social target on one side and an empty grid cage on the other side, and allowed to freely explore the chambers for 5 min.

The time spent and the trajectory taken in each chamber were recorded. The side containing the social target was called the target field, and the opposite side was called the nontarget field. Detailed procedures for preparation of subject animals, habituation, social targets, sociability, and social interaction tests, and handling of the behavior test room were carried out as described in a previous report[38,39].

**U-shaped two-choice field sociability test**. The U-shaped two-choice field sociability test was conducted as described previously[38,39]. The U-shaped two-choice field consisted of an open field (45 cm × 45 cm) partially partitioned with a wall (20 cm in width × 35 cm in height) to the central point to produce two symmetrical rectangular fields. A circular grid cage (12 cm in diameter × 33 cm in height) was placed on each side of the wall. For habituation, a subject mouse was allowed to freely explore the U-shaped field containing an empty grid cage on each side for 5 min. While the subject mouse was returned to its home cage for 2 min, a social target was loaded into one of the grid cages. Afterward, the subject mouse was again allowed to explore the U-shaped field for 5 min, and the time spent and trajectory taken were recorded. The side containing the social target was called the target field, and the opposite side was called the nontarget field.

**Sucrose-preference test**. The sucrose preference test (SPT) was carried out as described previously[69]. Mice were habituated to two water bottles for 7 days, followed by the presentation of two bottles containing a 1% sucrose solution for 24 h. The mice were then deprived of water and sucrose solution for 9 h (10 a.m.–7 p.m.). Beginning at 5 p.m., mice were singly housed in a cage provided with food but no water. At 7 p.m., each mouse was given two bottles, one containing water and the other containing a 1% sucrose solution. The positions of the water and sucrose bottles were changed at 8, 9, and 10 p.m. The amounts of water and sucrose consumed were measured by weighing the bottles. The sucrose preference (%) was calculated as the amount of sucrose solution consumed over the amount of water for 2 h (8–10 p.m.).

**Tail suspension test**. The tail suspension test was carried out as described previously[38,39]. Mice were individually suspended for 6 min by fixing their tails with adhesive tape to the ceiling of a rectangular box, leaving them 50 cm above the table surface. During the 6 min period, the cumulative immobility time was counted. Immobility was defined as the total time in which all limbs and the body did not move.

**Forced swim test**. The forced swim test was carried out as described previously[38,39]. Mice were placed for 6 min in a transparent Plexiglass cylinder (15 cm in diameter × 27 cm in height) containing water (a depth of 15 cm) at 24 °C. The cumulative immobility time was counted for the last 5 min. Immobility was defined as floating with all limbs motionless.

**Principal component analysis and $K$-means clustering**. The behavioral features of individual animals in multiple behavior tests were analyzed using $K$-means clustering, an unsupervised machine-learning algorithm that can portion individual data points into distinct subgroups without reference to known or labeled outcomes[70]. The principal component analysis (PCA) was used to transform the behavioral performance of individual animals in two behavior tests into a single eigenvector in a linear dimension, under the following conditions: $p < 0.05$ of Bartlett's test of sphericity, ≥0.5 of the Kaiser–Meyer–Olkin measure of sampling adequacy, ≥0.5 of communality, and ≥1.0 of the eigenvalues in the correlation matrix. The proportion of variance explained (PVE) for the dimension reductions in the present study was in the range of 73.4–86.3%. All individual variables in the behavioral tests and all transformed values were normalized into $z$-scores by calculating the mean ($\mu$) and standard deviation ($\sigma$). The $z$-scores were calculated as given below.

$$z = (x - \mu) / \sigma \qquad (1)$$

$K$-means clustering of the behavioral features of individual animals measured by three behavior tests were presented in three-dimensional matrices. The behavioral features of individual animals in four to five behavior tests and other physiological measurements of individual animals were visualized in a three-dimensional matrix for convenience after dimensional reduction of the TST × FST or SIT × SPT components. Prepared individual data points were randomly selected to find two centroids by computing the sum of the squared distance between data points. The two centroids used the following objective function ($J$) to subsequently assign each data point to the closest cluster center.

The objective function is:

$$J = \sum_{j=1}^{k} \sum_{i=1}^{n} \| x_i^{(j)} - c_j \|^2 \qquad (2)$$

$J$ = objective function, $k$ = number of clusters ($k = 2$, in this study), $n$ = number of subject groups, $x_i$ = value from behavioral tests, $c_j$ = centroid value of clusters.

Finally, the proportion (%) of partitioned data points in the two clusters was evaluated for each experimental group. PCA and $K$-means clustering were carried out using SPSS 25.0 software (IBM, Armonk, NY, USA).

**Statistical analysis.** Statistical analyses were conducted using GraphPad Prism 6 software (GraphPad Software, Inc., CA, USA). Two-sample comparisons were performed using a two-sided Student's $t$-test, and multiple comparisons were performed using one-way ANOVA and the Newman–Keuls post hoc test. All data were presented as mean ± SEM, and statistical differences were accepted at $p < 0.05$ unless otherwise indicated. The statistical details of the results of all figures are provided in Supplementary Data 4.

**Reporting Summary.** Further information on research design is available in the Nature Research Reporting Summary linked to this article.

## Data availability

All data collected or analyzed in this study are available within the article and Supplementary Information. Further information and requests for resources and reagents should be directed to and will be fulfilled by the corresponding author. The transcriptomic datasets are provided in Supplementary Data 2 and 3. The gene expression microarray data generated in this study has been deposited in the NCBI Gene Expression Omnibus (GEO) under the accession number GSE183624. Source data are provided with this paper.

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

## Acknowledgements
This research was supported by a grant (2021R1A2B5B02002245) from the Ministry of Science, ICT, and Future Planning, Republic of Korea.

## Author contributions
E.-H.L., J.-Y.P., and H.-J.K. carried out the experiments; E.-H.L. and P.-L.H. designed the experiments, performed the statistical analysis, and wrote the manuscript.

## Competing interests
The authors declare no competing interests.

## Additional information

**Peer review information** *Nature Communications* thanks Alfred Robison and the other anonymous reviewer(s) for their contribution to the peer review this work. Peer reviewer reports are available.

