## [Peer Review File · Nature Communications]

Reviewers' Comments:

Reviewer #1:

Remarks to the Author:

In this manuscript, Lee et al present a series of studies describing and characterizing an intervention in which they demonstrate that brief, repeated re-exposure to a stress can reverse depressive-like phenotypes elicited by repeated chronic stress. This is a very interesting result that will be of broad interest. They provide a robust parameterization of the behavioral side of this phenomenon, and of the role of corticosterone. Multiple timepoints, behavioral tasks and mouse lines are utilized. These figures are the strength of the manuscript. They then move into exploring potential cellular and molecular mechanisms that contribute, providing evidence that prelimbic cortical CaMKII signaling plays an important role. These aspects of the manuscript have more issues. I have some concerns regarding the immunohistochemical analyses and the siRNA-based knockdown experiments which are detailed below. The rationales for some of the experiments are also hard to follow, for example the brain regions to focus on based on the fos analysis, as well as the choice to pursue CaMKII function. Further, the DREADD experiments appear to lack key controls to ensure the actions are specific to CNO interactions with the DREADD rather than CNO actions on its own. This is particularly important given the lack of specificity observed across the various projections tested, raising the possibility that CNO is having a nonspecific action in these experiments. A number of other issues to consider to improve the manuscript are listed below.

- 1) The wording used to describe the phenomenon is very confusing. From the current title it is very difficult to discern what the manuscript is about. "Adaptively changed stress gains" does not seem to describe depressive phenotypes in my opinion. Similarly, "behavioral appraisal" does not capture repeated brief stress experiences. I suggest giving the manuscript a more specifically descriptive title, and more clearly describing the phenomenon under study in the abstract and introduction.
- 2) A more complete description of how the investigators define SDS susceptible and resilient populations needs to be provided.
- 3) Insufficient detail is provided in the figure legends and the methods for the antibodies (dilutions etc) utilized, and for the approaches used to validate the various siRNA knockdown experiments. Further, representative images need to be provided demonstrating the siRNA-induced knockdown of protein levels. Similarly, it's not clear how AVP and CRH levels were measured.
- 4) Insufficient rationale is provided for the choice of brain regions to pursue fos imaging in. It's also not clear what the marker for glutamate neurons is in this experiment. Further, based on the data in figure 4, it is not clear why the subiculum was not pursued further.
- 5) The panel organization of figure 5 is oddly arranged and hard to follow. P-t in this figure is an example of one of the siRNA experiments where it's not clear what the validation bar graph refers to, and is in need of a representative image. Panel I, like many of the other representative images shown, is not convincing in terms of potential differences described. The quantification in panels j-l at first glance appears very robust, however it is individual cells, which is not a useful way to present these data. Data variation needs to be shown across animals rather than individual cells, and statistical analysis needs to be across animals as well.
- 6) The representative images shown in figure 6 do not convey observable differences in signal relative to the quantification shown. As in the previous figure comments, I am concerned that the method of quantifying these data, which appears to be across large numbers of cells rather than across animals, is not appropriate. Further, no rationale is provided for the switch to focusing on CaMKII function, which did not appear to emerge from the transcriptional analysis data in figure 5, nor of the specific GABA subunits further investigated.
- 7) The WGA CRE utilized has the potential for both retrograde and anterograde travel. It's not clear the the potential anterograde aspects of this mechanism were considered in the design.

Reviewer #2:

Remarks to the Author:

The manuscript "Behavioral appraisal by implementing a short sequence of stress resolves adaptively changed stress gains" uses mouse behavior, pharmacology, genomics, biochemistry, molecular biology, and chemogenetic circuit manipulation to demonstrate a clear mechanism of

PFC neuronal activity underlying the antidepressant effects of repeated weak stress in mouse models of chronic-stress-induced depression-like behavior. This is a genuine tour de force study that is overwhelmingly comprehensive in its behavioral and molecular approaches. It broaches new territory in uncovering a cellular and molecular mechanism (CaMKII-mediated excitatory activity of PFC projection neurons) underlying antidepressant results of repeated weak stress exposure. Some examples of the efforts to produce a robust and reliable study include: the use of multiple chronic stress paradigms (chronic restraint and chronic social defeat); the use of multiple depression-related behavioral outcomes (sociability, sucrose preference, TST, FST, social interaction); the comprehensive measures and manipulations of CORT levels, including many time points and doses; the extensive validation of gene targets uncovered through sequencing; the multiple approaches (pharmacological and genetic) to establishing causal connections between a target (CaMKII) and a behavioral outcome; and the general and circuit-level chemogenetic approaches to connecting PFC function to behavioral outcome. At each of these steps, the authors use multiple appropriate controls, and all statistical analyses are correct. The manuscript is well written and the interpretations are justified and well-informed by the literature. Frankly, this is one of the finest manuscripts I have ever reviewed. It is exciting, will have strong impact on the field, and is sure to garner many citations and inspire a range of follow up studies. My only very minor criticism is that the final model (Fig 10) would be improved if small titles were placed above each of the images (b, c, and e) explaining the state (naïve, after chronic stress, and after repeated weak stress, respectively). Aside from this, I have no suggested changes, and would strongly recommend the manuscript for acceptance and publication in its current form.

AJ Robison
Associate Professor of Physiology
Michigan State University

Reviewer #3:

Remarks to the Author:

Lee and colleagues have presented an impressively comprehensive set of experiments in support of the idea that exposure to weak stressors can alleviate aspects of dysfunction induced by chronic stress. The authors showed that repeated daily treatment with 5-min restraint or low dose glucocorticoids has "antidepressive" effects via PL activation. Using a series of in vivo and ex vivo methods, they have implicated the PLdBNST pathway in mediation of this effect.

The studies presented provide a diverse and convincing argument for a circuit specific mechanism mediating the ability for mild stress to counteract effects of chronic stressors, and have potential for great impact in the field. I offer suggestions to further improve the manuscript:

- Being that many other studies have implicated the IL PFC in stress related behaviors and have shown that manipulating IL can alter CORT/ stress, it was surprising that there was no mention of IL or experimental manipulation/ cfos expression data, etc. While I do not think additional experiments to manipulate IL are necessary, it may be useful to quantify cfos expression in IL (assuming the tissue is still around) for comparison with PL. At the very least, there should be a discussion of the potential role for IL in the discussion section, as this would be important for the larger picture presented here.
- There have been an increasing number of arguments in the field for not referring to behavior in the rodent assays used here as "depression". The authors should revise the manuscript accordingly. (For example, a discussion on construct validity of FST: PMC5518600).
- The figures could be improved for clarity. Currently the text is very small and hard to read, and the panels seem cluttered. Revision of layouts/ fonts would facilitate a readers' ability to interpret the data presented.

Point-by-point Responses to the reviewers' comments

We appreciate the reviewer's very constructive and insightful comments on the manuscript [NCOMMS-20-40200]. We put our best efforts including new experiments to address all of the issues and concerns raised by the reviewers. We believe the manuscript has been greatly improved through this revision. Here are the point-by-point responses to the reviewers' comments and concerns.

REVIEWER COMMENTS

Reviewer #1 (Remarks to the Author):

In this manuscript, Lee et al present a series of studies describing and characterizing an intervention in which they demonstrate that brief, repeated re-exposure to a stress can reverse depressive-like phenotypes elicited by repeated chronic stress. This is a very interesting result that will be of broad interest. They provide a robust parameterization of the behavioral side of this phenomenon, and of the role of corticosterone. Multiple timepoints, behavioral tasks and mouse lines are utilized. These figures are the strength of the manuscript. They then move into exploring potential cellular and molecular mechanisms that contribute, providing evidence that prefrontal cortical CaMKII signaling plays an important role. These aspects of the manuscript have more issues.

I have some concerns regarding the immunohistochemical analyses and the siRNA-based knockdown experiments which are detailed below. The rationales for some of the experiments are also hard to follow, for example the brain regions to focus on based on the fos analysis, as well as the choice to pursue CaMKII function.

Further, the DREADD experiments appear to lack key controls to ensure the actions are specific to CNO interactions with the DREADD rather than CNO actions on its own. This is particularly important given the lack of specificity observed across the various projections tested, raising the possibility that CNO is having a nonspecific action in these experiments. A number of other issues to consider to improve the manuscript are listed below.

→ The reviewer raises a legitimate concern that CNO could action on its own. We carried out a new experiment to test if CNO treatment changes stress-induced depressive phenotypes. Repeated injection of CNO at 3 mg/kg/injection, the highest dose we used in the present study, in CRST mice did not affect stress-induced depressive-like behavior. We include this result in Extended figure 9 of the revised manuscript.

(1) The wording used to describe the phenomenon is very confusing. From the current title it is very difficult to discern what the manuscript is about. "Adaptively changed stress gains" does not seem to describe depressive phenotypes in my opinion. Similarly, "behavioral appraisal" does not capture repeated brief stress experiences. I suggest giving the manuscript a more specifically descriptive title, and more clearly describing the phenomenon under study in the abstract and introduction.

→ We narrow down the scope of behavioral appraisal and stress gains in the

abstract and introduction. We also slightly modify the title in the revised manuscript.

(2) A more complete description of how the investigators define SDS susceptible and resilient populations needs to be provided.

→ We describe and provide the behavioral data to select SDS susceptible and resilient individuals in Extended Data Figure 2, which includes representative trackings (Extended Figure 2a,b), distribution of individuals (Figure 1k), the time spent in the interaction zone and corner zone (Extended Figure 2c,d), and sociability scores of SDS susceptible and resilient individuals (Extended Figure 2e,f).

(3) Insufficient detail is provided in the figure legends and the methods for the antibodies (dilutions etc) utilized, and for the approaches used to validate the various siRNA knockdown experiments. Further, representative images need to be provided demonstrating the siRNA-induced knockdown of protein levels. Similarly, it's not clear how AVP and CRH levels were measured.

→ We describe the details of the antibodies utilized and of siRNA knockdown experiments in the Supplementary Methods.

We provide new immunohistochemical images showing the knockdown of GR (Figure 5p) and CaMKIIa (Figure 6p) in the PL regions injected with siRNA-GR or siRNA-CaMKIIa. We provide also the qPCR data showing the siRNA-mediated knockdown of transcript levels of GR (Figure 5q), CaMKIIa (Figure 6m), and Erk1 and Erk2 (Figure 6n,o) in the injection sites.

The transcript levels of CRH and AVP in the PVN were assessed 72 hours after the injection of siRNA-GR in the PL (Figure 5s).

(4) Insufficient rationale is provided for the choice of brain regions to pursue fos imaging in. It's also not clear what the marker for glutamate neurons is in this experiment. Further, based on the data in figure 4, it is not clear why the subiculum was not pursued further.

→ We rewrite the pointed part by the reviewer of the Results and add a transition statement for the switch to focusing on *c-Fos*-based approach and the choice of the PL to investigate further.

As the reviewer pointed, the subiculum was distinctively activated by repeated 5-min restraint. Although we have focused on the PL in the present study, it is possible that the subiculum has a role in RS5 effects. We include a comment on the possible role of the subiculum in RS5 effects in the Discussion section of the revised manuscript.

We used anti-GLU-4 as a glutamatergic neuronal marker. GLU-4 is a monoclonal anti-glutamate antibody produced in mouse cell, and it detects L-glutamate when immobilized with glutaraldehyde (G9282, Sigma-Aldrich).

(5) The panel organization of figure 5 is oddly arranged and hard to follow. P-t in this figure is an example of one of the siRNA experiments where it's not clear what the validation bar graph refers to, and is in need of a representative image. Panel I, like many of the other representative images shown, is not convincing in terms of

potential differences described. The quantification in panels j-l at first glance appears very robust, however it is individual cells, which is not a useful way to present these data.

→ We reorganize the layout of Figure 5 and provide new quantification data for the expression levels of GR and Fkbp5 assessed across brain sections of animals (Figure 5j-l), and new immunohistochemical images showing siRNA-mediated knockdown of GR in the injection site (Figure 5p).

(6) The representative images shown in figure 6 do not convey observable differences in signal relative to the quantification shown. As in the previous figure comments, I am concerned that the method of quantifying these data, which appears to be across large numbers of cells rather than across animals, is not appropriate. Further, no rationale is provided for the switch to focusing on CaMKII function, which did not appear to emerge from the transcriptional analysis data in figure 5, nor of the specific GABA subunits further investigated.

→ We provide new immunohistochemical images showing p-CaMKII α and GR expression in the PL (Figure 6f) and new quantification data for the expression levels of p-CaMKII α and GR assessed across brain sections of animals (Figures 6g-i).

We rewrite the pointed part by the reviewer of the Results and add a transition statement for the switch to focusing on CaMKII function and GABA subunits.

(7) The WGA CRE utilized has the potential for both retrograde and anterograde travel. Its not clear the potential anterograde aspects of this mechanism were considered in the design.

→ The reviewer raises an important question regarding the retrograde and anterograde property of WGA-CRE. We applied following approaches to examine potential anterograde and retrograde effects of WGA-CRE.

First, we carefully examined whether and how DREADD-mediated inhibition of the three PL outputs (PL→NAcc, PL→dBNST, and PL→BLA) changed RS5-induced *c-Fos* expression in PL neurons and their target sites. We found that CNO injection in mice with the DREADD-hM3D(Gq) in the PL→NAcc, PL→dBNST, or PL→BLA circuits reduced RS5-induced *c-Fos* expression in the NAcc, dBNST and BLA, respectively, but not in the PL and PL target regions (Figure 9c-f; Figure 9n-s).

Second, we also carried out a new experiment to transduce hM3D(Gq) stimulatory DREADD in the PL→dBNST circuit, and found that stimulation of the PL→dBNST by injection of CNO strongly increased *c-Fos* expression in the dBNST, but no CNO-dependent increase of *c-Fos* expression in the PL, NAcc, and BLA (Extended Figure 11).

Third, we carried out a new optogenetic experiment demonstrating that optic inhibition of PL→NAcc neurons in CRST mice that received RS5 treatment exhibited depressive-like behavior (Extended Figure 12). This optogenetic data is consistent with the result that DREADDs-mediated inhibition of PL→NAcc neurons in CRST mice that received RS5 treatment produced depressive-like behaviors (Figure 9u-x).

These results are included in the main figure Fig. 9, Extended figure 11 and Extended figure 12 in the revised manuscript.

Reviewer #2 (Remarks to the Author):

The manuscript “Behavioral appraisal by implementing a short sequence of stress resolves adaptively changed stress gains” uses mouse behavior, pharmacology, genomics, biochemistry, molecular biology, and chemogenetic circuit manipulation to demonstrate a clear mechanism of PFC neuronal activity underlying the antidepressant effects of repeated weak stress in mouse models of chronic-stress-induced depression-like behavior. This is a genuine tour de force study that is overwhelmingly comprehensive in its behavioral and molecular approaches. It broaches new territory in uncovering a cellular and molecular mechanism (CaMKII-mediated excitatory activity of PFC projection neurons) underlying antidepressant results of repeated weak stress exposure. Some examples of the efforts to produce a robust and reliable study include: the use of multiple chronic stress paradigms (chronic restraint and chronic social defeat); the use of multiple depression-related behavioral outcomes (sociability, sucrose preference, TST, FST, social interaction); the comprehensive measures and manipulations of CORT levels, including many time points and doses; the extensive validation of gene targets uncovered through sequencing; the multiple approaches (pharmacological and genetic) to establishing causal connections between a target (CaMKII) and a behavioral outcome; and the general and circuit-level chemogenetic approaches to connecting PFC function to behavioral outcome. At each of these steps, the authors use multiple appropriate controls, and all statistical analyses are correct. The manuscript is well written and the interpretations are justified and well-informed by the literature. Frankly, this is one of the finest manuscripts I have ever reviewed. It is exciting, will have strong impact on the field, and is sure to garner many citations and inspire a range of follow up studies.

My only very minor criticism is that the final model (Fig 10) would be improved if small titles were placed above each of the images (b, c, and e) explaining the state (naive, after chronic stress, and after repeated weak stress, respectively). Aside from this, I have no suggested changes, and would strongly recommend the manuscript for acceptance and publication in its current form.

→ Thanks. We add explaining titles above the diagrams (b, c, and e) of Figure 10.

Reviewer #3 (Remarks to the Author):

Lee and colleagues have presented an impressively comprehensive set of experiments in support of the idea that exposure to weak stressors can alleviate aspects of dysfunction induced by chronic stress. The authors showed that repeated daily treatment with 5-min restraint or low dose glucocorticoids has "antidepressive" effects via PL activation. Using a series of in vivo and ex vivo methods, they have

implicated the PLdBNST pathway in mediation of this effect.

The studies presented provide a diverse and convincing argument for a circuit specific mechanism mediating the ability for mild stress to counteract effects of chronic stressors, and have potential for great impact in the field. I offer suggestions to further improve the manuscript:

(1) Being that many other studies have implicated the IL PFC in stress related behaviors and have shown that manipulating IL can alter CORT/ stress, it was surprising that there was no mention of IL or experimental manipulation/ cfos expression data, etc. While I do not think additional experiments to manipulate IL are necessary, it may be useful to quantify cfos expression in IL (assuming the tissue is still around) for comparison with PL. At the very least, there should be a discussion of the potential role for IL in the discussion section, as this would be important for the larger picture presented here.

→ As the reviewer pointed, the IL and PL constitute the core of the medial prefrontal cortex that regulates stress-related responses and emotional behaviors. We provide the detailed data for *c-Fos* expression in the PL and IL induced by short-term stress and low-dose CORT treatment conditions in Fig. 4b and Supplementary Table 1. We observed that the IL also had increased *c-Fos* expression after repeated weak stress (Fig. 4a,b). Therefore, it is possible that the IL has a role in stress-related depressive behaviors and RS5 effects. We add a discussion on the potential role of the IL in RS5 effects in the discussion section.

(2) There have been an increasing number of arguments in the field for not referring to behavior in the rodent assays used here as "depression". The authors should revise the manuscript accordingly. (For example, a discussion on construct validity of FST: PMC5518600).

→ As the reviewer pointed, there has been arguments that the FST behavior does not model human depression, but it is a part of stress coping strategy. We agree with this view. In the present study, we applied not only the FST, but also the TST, sociability, and sucrose preference test to read out animal behaviors. Probably the TST behavior might be taken as a part of stress coping strategy in parallel to the FST, and behavioral changes in the sociability and sucrose preference test could be interpreted as such, although the latter could raise other questions and debates. We include a discussion on the possible interpretation of the FST behavior as a stress coping strategy in the Discussion section of the revised manuscript.

(3) The figures could be improved for clarity. Currently the text is very small and hard to read, and the panels seem cluttered. Revision of layouts/ fonts would facilitate a readers' ability to interpret the data presented.

→ Thanks for the comments. We revise and change the layouts and font sizes in almost all figures to improve readability of figures in the revised manuscript.

Reviewers' Comments:

Reviewer #1:

Remarks to the Author:

The authors provide a strong revision of the manuscript. I have two remaining concerns:

- 1) I continue to think that the title of the manuscript is too speculative relative to the results shown, and should be more specifically descriptive of the work done. In particular, it is not clear what "Behavioral appraisal" means, but it appears to be a speculative interpretation of mouse cognition that exceeds the data.
- 2) The authors still need to provide their explicit criterion for the binning of animals into susceptible and vulnerable populations. Example behavioral data are shown, but no group splitting approach description is provided, other than stating "on the basis of a sociability ratio". A clear specific description of how animals are binned into the two groups is required. How exactly is the sociability ratio utilized?

Reviewer #2:

Remarks to the Author:

The already outstanding manuscript has been further improved by the extensive revisions.

AJ Robison

Reviewer #3:

Remarks to the Author:

The authors have adequately addressed concerns and suggestions proposed in my original review.

Point-by-Point Responses to the reviewers' comments

We appreciate the reviewer's constructive comments on the manuscript [NCOMMS-20-40200B]. We addressed the remaining concerns raised by the reviewer 1. We believe the manuscript has been greatly improved through these revisions. Here are the point-by-point responses to the reviewers' comment and concern.

REVIEWER COMMENTS

Reviewer #1 (Remarks to the Author):

The authors provide a strong revision of the manuscript. I have two remaining concerns:

1) I continue to think that the title of the manuscript is too speculative relative to the results shown, and should be more specifically descriptive of the work done. In particular, it is not clear what "Behavioral appraisal" means, but it appears to be a speculative interpretation of mouse cognition that exceeds the data.

→ We replace it with a new title in the revised manuscript.

2) The authors still need to provide their explicit criterion for the binning of animals into susceptible and vulnerable populations. Example behavioral data are shown, but no group splitting approach description is provided, other than stating "on the basis of a sociability ratio". A clear specific description of how animals are binned into the two groups is required. How exactly is the sociability ratio utilized?

→ We provide the procedure for the selection of susceptible and resilient groups in the Methods section of "Chronic social defeat stress (CSDS)".